# Decision Trees with Short Explainable Rules

**Victor F. C. Souza**
Departamento de Informática
Pontifical Catholic University of Rio de Janeiro
Rio de Janeiro, RJ - Brazil
`vfsouza@inf.puc-rio.br`

**Ferdinando Cicalese**
Department of Computer Science
University of Verona
Verona - Italy
`ferdinando.cicalese@univr.it`

**Eduardo Sany Laber**
Departamento de Informática
Pontifical Catholic University of Rio de Janeiro
Rio de Janeiro, RJ - Brazil
`laber@inf.puc-rio.br`

**Marco Molinaro**
Microsoft Research & Pontifical Catholic University of Rio de Janeiro
`mmolinaro@microsoft.com`

## Abstract

Decision trees are widely used in many settings where interpretable models are preferred or required. As confirmed by recent empirical studies, the interpretability/explainability of a decision tree critically depends on some of its structural parameters, like size and the average/maximum depth of its leaves. There is indeed a vast literature on the design and analysis of decision tree algorithms that aim at optimizing these parameters.

This paper contributes to this important line of research: we propose as a novel criterion of measuring the interpretability of a decision tree, the sparsity of the set of attributes that are (on average) required to explain the classification of the examples. We give a tight characterization of the best possible guarantees achievable by a decision tree built to optimize both our new measure (which we call the explanation size) and the more classical measures of worst-case and average depth. In particular, we give an algorithm that guarantees $O(\ln n)$-approximation (hence optimal if $P \neq NP$) for the minimization of both the average/worst-case explanation size and the average/worst-case depth. In addition to our theoretical contributions, experiments with 20 real datasets show that our algorithm has accuracy competitive with CART while producing trees that allow for much simpler explanations.

## 1 Introduction

Machine learning models and algorithms appear more and more frequently in systems that make decisions with an impact in our lives. Thus, it is highly desirable that the output of these methods

36th Conference on Neural Information Processing Systems (NeurIPS 2022).

are interpretable so that we can use them more comfortably or, eventually, question its applicability [13].[1]

Decision trees are used in many settings as a tool to provide explainability. However, their explainability greatly depends on the depths of its leaves, as empirically demonstrated by [46]. In fact, based on empirical data from a survey with 98 questions answered by 69 respondents, the authors conclude that "question depth" (the depth of the deepest leaf that is required when answering questions about a tree) turns out to be the most important parameter. Essentially, users prefer trees where the information about the most common items are given at the top of the tree. Minimizing the average/worst-case depth indeed has been a classic goal for decision tree algorithms (see the Related Work section below).

However, another very important component for explainability is having decision rules that are sparse, namely, that use as few different attributes as possible to classify an object or make a prediction. For decision trees, this means that the path to any given leaf should test only a small number of different attributes. Figure 1 shows two trees for the `Sensorless` dataset [7] with similar accuracy. While the trees have the same size, the rightmost one gives much more concise classification rules. As a spoiler, the left tree was constructed using `CART` and the right one using the algorithm we propose in this paper. To exemplify their differences, let us consider the leaves that are marked in the figure with a thick rectangle. Both are assigned the same class and cover the maximum number of examples in the training set (approximately 11.500 example each). Despite this similar behaviour, the explanations for their classifications are quite different (the $D_i$'s denote the attributes):

$$
\begin{array}{ll}
\texttt{CART:} & D_{11} \in \texttt{[-0.11, 0.07]} \text{ AND} \\
& D_9 > \texttt{-0.01} \text{ AND} \\
& D_{10} \leq \texttt{0.03} \\
\texttt{New algorithm:} & D_{11} \in \texttt{[-0.01, 0.03]}
\end{array}
$$

Define the explanation size of a leaf as the number of distinct attributes tested on the path from the root to this leaf. The above example shows that having trees whose leaves have small explanation size yields significantly simpler (hence easier to interpret/explain) rules. However, to the best of our knowledge there is no prior work considering decision trees optimized with respect to the explanation size.

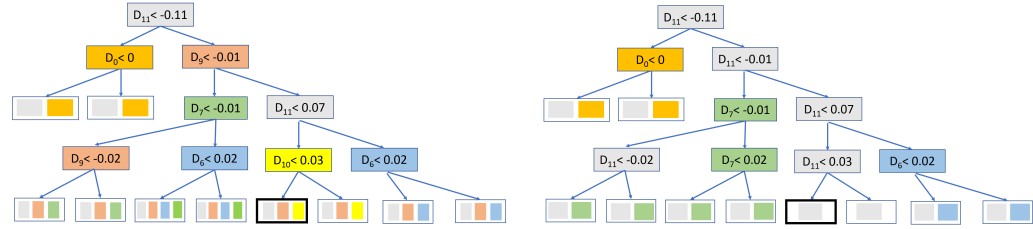

Figure 1: The left and right decision trees are built, respectively, by `CART` and the algorithm proposed in this paper for the `Sensorless` dataset. The maximum depth was set to 4. Internal nodes associated with the same attribute have the same color. The colors inside a leaf indicate the attributes that are used to obtain its classification.

**Our contributions.** In this work we propose the explanation size as a novel measure to capture the interpretability of a decision tree. In particular, we initiate a principled study of decision trees with small explanation size by focusing on: (i) the trade-off that optimizing this criterion imposes on other desirable metrics (e.g., average/worst-case depth); (ii) the design of efficient algorithms for building such trees that guarantee good performance in practice.

Our first result is that it is possible to essentially obtain a best-of-both worlds in terms of average/worst-case depth and explanation size: We show that there is always a binary decision tree that has the smallest average explanation size possible but also has average depth not much larger than that of the

---

[1]We use the words interpretability and explainability in a broad sense; for a detailed discussion of the different concepts related to the topic (see, for example, [41]).

optimal tree for the latter metric (Item 2 of Theorem 1). The same result also holds when considering worst-case explanation size and depth (Item 1 of Theorem 1). Remarkably, the latter (worst-case bound) turns out to be tight. Theorem 2 shows it matches a necessary trade-off between optimizing explanation size and depth, i.e., improving the bound on the depth in Theorem 1 can only be attained at the cost of a logarithmic factor loss in the explanation size.

Despite having strong theoretical guarantees, the construction to obtain these trees is too wasteful to be used in practice. Thus, our second contribution is an algorithm that still yields a tree that provably approximates both optimal average/worst-case explanation size and depth (Theorem 3) but has enough flexibility that it can be employed to obtain a good performance in practice. To demonstrate the applicability of our proposal, we compare it against `CART` [11], a quite popular method to build decision trees, on 20 real classification datasets. Our method leads to much more (resp. more) interpretable trees in terms of explanation size (resp. average depth), while having a performance similar to `CART` in terms of accuracy and speed.

## 2 Related work

Our work can be connected with an active line of research that proposes interpretable models for machine learning [13], in particular more interpretable rule-based models (e.g. decision lists, sets, and trees) [37, 30, 21, 9]. Our interpretability metric is closely related to rule sizes considered in the rule learning literature, e.g., [31, 47]. In addition, we can relate our work with those from the vast literature of methods with provable guarantees that are designed to build decision trees of "low complexity" (e.g. depth of leaves, number of nodes, etc.) [23, 22, 18, 8, 24].

**More interpretable rule-based models.** There is a body of work that aims to understand what makes a rule model more comprehensible via experiments with end users [4, 29, 46]. The paper [4] compares the comprehensibility of classifiers that are learned by decision trees and rule-learning algorithms, based on subjective comparisons of classifier pairs by 100 Computer Science students. They conclude that decision trees are more comprehensible. In [29], based on experiments with business students, it is concluded that decision tables are more interpretable than decision trees and propositional "if-then" rules. One potential limitation of this study is that these students had no experience with the representation formats, so it is not clear whether this conclusion can be extended for more experienced users. The work of [46] reports a survey with 69 respondents that was carried out to understand what makes a decision tree more interpretable. Among their findings is that the depth of the leaves required by users during the experiments had one of the biggest and most consistent impact on the usability of decision trees across multiple tasks (classify, explain, validate, and discover).

There are some recent works that try to optimize interpretabilty metrics when building rule-based models [37, 10, 56, 30, 40, 9]. We briefly discuss those that focus on decision trees. The paper [30] contends that splits that have at least one of its parts/child nodes being class-homogeneous (roughly this means that most examples have the same class) are important for interpretability, and propose a splitting criterion that tries to balance this homogeneity and the depth of the leaves. In contrast to our work, no provable guarantees are provided. The works [10, 56] employ Integer Linear Programming to build trees of a given maximum depth, while [40] employs Dynamic Programming techniques to develop an optimization framework that allows the construction of decision trees with few leaves that optimize a variety of objective functions such as F-score and `AUC`. These methods, while interesting, are much more complex to implement than ours and they do not run in polynomial time, which may compromise their application on large datasets.

**Decision trees with provable guarantees.** There is a vast literature dedicated to the problem of building decision trees with "low complexity", where the complexity of a tree can be measured in different ways, including worst-case/average leaf depth and number of nodes, among others [12]. It is known that building a decision tree that minimizes the worst-case or average leaf depth, among those that fit the training data, does not admit an $o(\log n)$ approximation unless P=NP [16, 36]. When the goal is minimizing the number of nodes, the problem is even harder to approximate [3, 26]. Regarding upper bounds, several algorithms with the optimal (within constants) $O(\log n)$-approximation are known for minimizing the worst-case and average depth [23, 22, 17, 18, 8, 24, 32, 39, 44]. What distinguishes each method is the generality of its guarantee; for example, some consider scenarios

that include tests with noisy outcomes [32], while others consider items/examples with non-uniform weights and tests with non-uniform costs [18, 44].

We remark that the method we propose here also allows the use of non-uniform weights on the examples, which can be used, for instance, to prioritize models that yield simpler explanations for some classes of particular interest, by setting high weights for examples in these classes. The key difference between our method and the existing ones is that ours is the only one that provides theoretical guarantees on the explanation size.

## 3 Model

We describe the model used throughout for the theoretical analyses. For that, we adopt a terminology similar to that employed by some closely related works [1, 18]. On an instance $I$, there is a set of ordinal attributes $A$ and a set of objects $O$, where each object $o \in O$ is described by the value it takes on each attribute $a \in A$; we denote this value by $a(o)$. Each object $o$ also has a class $c(o)$ in some set $C$. In order to classify an object $o$, we are allowed to use threshold tests of the form "Is $a(o) < t$?" for some attribute $a$ and threshold value $t$.

A (threshold) decision tree $T$ is a rooted tree where each internal node $\nu$ is associated with a threshold test "Is $a(o) < t$?" and the edges from a node to its children are associated to the two possible outcomes "$a(o) < t$" and "$a(o) \geq t$". We also refer to this test by the attribute/threshold pair $(a, t)$. Each leaf $\ell$ of $T$ is associated with a class in $C$. Given a decision tree $T$, we say that an object $o$ reaches a node $\nu$ of $T$ if it agrees with all outcomes associated with the path from the root of $T$ to $\nu$. For each $o \in O$, we use $\ell(o)$ to denote the unique leaf reached by object $o$. Finally, we consider the exact classification model, namely a decision tree must correctly classify all objects of the instance, i.e., each object $o \in O$ reaches some leaf associated to its correct class $c(o)$.

We now formalize the measures of interpretability that were mentioned in the introduction.

**Depth and explanation size of a tree.** We start recalling the classical notions of worst-case and average tree depth. Given a decision tree $T$, the depth of a leaf $\ell$ is the number of tests/internal-nodes on the path from the root of $T$ to $\ell$, and is denoted by depth($\ell$). The worst-case depth of the tree $T$ is obtained by considering the maximum depth over its leaves, namely

$$\text{depth}_{wc}(T) := \max_{\ell \in \text{leaf}(T)} \text{depth}(\ell).$$

To measure the average depth of the tree, in addition to the datum above, as part of the input each object $o$ has a non-negative weight $w(o) \in \mathbb{R}_+$ indicating its likelihood/importance. Letting $w(\ell)$ be the sum of the weights of all objects that reach leaf $\ell$, the average depth of the tree $T$ is then the weighted sum of the depth of its leaves, namely

$$\text{depth}_{avg}(T) := \sum_{\ell \in \text{leaf}(T)} w(\ell) \cdot \text{depth}(\ell).$$

We now consider the novel measures of quality/interpretability of a tree using the notion of explanation size. The explanation size of a leaf $\ell$, denoted by expl($\ell$), is the number of different attributes on the tests on the path from the root of $T$ to $\ell$. As an example, the explanation size of the leaf marked with a thick rectangle on the left tree of Figure 1 is 3. The worst-case and average explanation size of a tree $T$ are then given as before by the largest and the weighted sum of the explanation sizes of its leaves, respectively:

$$\text{expl}_{wc}(T) = \max_{\ell \in \text{leaf}(T)} \text{expl}(\ell),$$

$$\text{expl}_{avg}(T) = \sum_{\ell \in \text{leaf}(T)} w(\ell) \cdot \text{expl}(\ell).$$

Our goal is to obtain trees with as small worst-case/average depth and explanation size as possible. We use $\text{depth}_{wc}^* = \text{depth}_{wc}^*(I)$ to denote the smallest possible worst-case depth of a decision tree that solves instance $I$, and define the optimal values $\text{expl}_{wc}^*$, $\text{depth}_{avg}^*$, and $\text{expl}_{avg}^*$ analogously.

# 4 Tradeoff between depth and explanation size

Our goal is to obtain trees that simultaneously have both small worst-case/average depth and explanation size. However, is this even possible? It is conceivable that in order to obtain trees with small depth, one may be required to use several different attributes along the paths to effectively classify the objects; but this would make the tree have a large explanation size. Conversely, in a tree with small explanation size, the few attributes along a path may need to be used many times in order to correctly classify the objects, leading to a large tree depth.

Perhaps surprisingly, we show that there are trees that simultaneously have optimal explanation size and almost optimal depth.

**Theorem 1.** *Given an instance of the classification problem, the following holds:*

1. *(Worst-case metrics) There exists a binary tree $T$ for which simultaneously*
    - $\mathrm{expl}_{wc}(T) = \mathrm{expl}^*_{wc}$
    - $\mathrm{depth}_{wc}(T) \leq 2\,\mathrm{depth}^*_{wc} + \log n.$

2. *(Average metrics) There exists a binary tree $T$ for which simultaneously*
    - $\mathrm{expl}_{avg}(T) = \mathrm{expl}^*_{avg}$
    - $\mathrm{depth}_{avg}(T) \leq 2\,\mathrm{depth}^*_{avg} + W \log n.,$

*where $W$ is the sum of the weights of the object in the instance.*

We remark that these additive bounds on the worst-case and average depth imply $O(\log n)$ multiplicative approximations as well.

**Observation 1.** *The bounds from Theorem 1 imply*

$$\mathrm{depth}_{wc}(T) \leq \frac{3 \log n}{\log c} \cdot \mathrm{depth}^*_{wc}$$
$$\mathrm{depth}_{avg}(T) \leq 3 \log n \cdot \mathrm{depth}^*_{avg},$$

*where $c$ is the number of classes.*

While the proof of Theorem 1 is deferred to Appendix B, we give here the main ideas behind it. We will consider here only the worst-case metric (Item 1), since the proof is simpler and more transparent.

To show the existence of our desired tree we make use of multiway trees, i.e., a decision tree where multiway tests are used rather than threshold tests. A multiway test associated with attribute $a$ splits the objects based on all possible values of this attribute. As an example, if an attribute $a$ takes 5 distinct values for the objects in the instance and we use $a$ at the root of a multiway tree, then the root will have 5 children.

The starting point for the construction of the tree in Theorem 1 is the equivalence between optimal binary trees and optimal multiway trees in terms of worst-case explanation size. While this is formally proved in Lemmas 2 and $\overline{3}$, for an intuitive view of this equivalence first notice that there is a multiway tree $M^*$ that is simultaneously optimal in terms of worst-case depth and worst-case explanation size, since each attribute only needs to be used once in a path. Also, this optimal multiway tree $M^*$ has worst-case explanation size (equivalently worst-case depth) at most that of the best binary tree, namely $\mathrm{depth}_{wc}(M^*) = \mathrm{expl}_{wc}(M^*) \leq \mathrm{expl}^*_{wc}$, since intuitively multiway tests are more informative than binary tests. Conversely, we can transform an optimal multiway tree $M^*$ into a binary decision tree $T$ by simulating each multiway test on an attribute $a$ by using multiple threshold tests "Is $a(o) < t$" with varying $t$ (but same attribute $a$). Since explanation size only counts the number of distinct attributes used along a path, the tree $T$ so created has exactly the same explanation sizes as $M^*$, and hence $\mathrm{expl}_{wc}(M^*) = \mathrm{expl}_{wc}(T) \geq \mathrm{expl}^*_{wc}$. Thus, we have the equivalence $\mathrm{expl}_{wc}(M^*) = \mathrm{expl}^*_{wc}$.

To prove Theorem 1, we start with the optimal multiway tree $M^*$ and convert it into a binary tree, as above. However, the conversion in the previous paragraph is not enough: while it preserves the explanation size, it may greatly increase the depth of the leaves when using multiple threshold tests to simulate a multiway test (possibly yielding depth $\gg \mathrm{depth}^*_{wc}$). The key idea is to use a much more efficient simulation that is based on alphabetic codes, a classic notion from coding theory [28].

The following result from [2, Chp. 2, p. 341], rephrased in the terminology of decision trees, gives a sufficient condition for the existence of such codes with prescribed code-lengths $d_i$'s.

**Lemma 1.** *Consider an instance of the classification problem with $n$ objects but only 1 attribute. Then for any positive integers $d_1, \ldots, d_n$ such that $\sum_{j=1}^{n} 2^{-d_i} \leq \frac{1}{2}$, there exists a binary threshold decision tree with $n$ leaves at depths $d_1, \ldots d_n$ such that the $i$-th object reaches the leaf at depth $d_i$.*

Then for each node $\nu$ of $M^*$ (corresponding to an attribute $a$, and with children $ch_1, ch_2, \ldots$), we consider all objects that reach a child $ch_i$ as a "single object" (with the corresponding value in attribute $a$) and applying the previous lemma we replace $\nu$ by a binary tree $A$ where the leaf corresponding to the objects in $ch_i$ end up in a leaf at depth (in $A$) $\ell_i = \lceil \log \frac{n(\nu)}{n(ch_i)} \rceil + 1$, where $n(node)$ is the number of objects that reach a given node in $M^*$. The final tree obtained, call it $T$, still has the same explanation sizes as $M^*$, so $\mathrm{expl}_{wc}(T) = \mathrm{expl}^*_{wc}$. Moreover, in terms of worst-case depth, any root-to-leaf path $P$ in $T$ has a corresponding path $P_{M^*} = \nu_0, \nu_1, \nu_2, \ldots$ in $M^*$, and the length of $P$ is at most

$$\sum_{i=0}^{|P_{M^*}|-1} \left( \left\lceil \log \frac{n(\nu_i)}{n(\nu_{i+1})} \right\rceil + 1 \right) \;\leq\; \log n + 2 \cdot [\text{length of } P_{M^*}].$$

By looking at the longest such path we get

$$\mathrm{depth}_{wc}(T) \;\leq\; \log n + 2\,\mathrm{depth}_{wc}(M^*) \;\leq\; \log n + 2\,\mathrm{depth}^*_{wc},$$

where the last inequality holds because of the optimality of $M^*$ with respect to worst-case depth. This gives Item 1 of Theorem 1.

The average-case part of the theorem (Item 2) uses similar ideas, but in addition relies on entropy-based calculations to argue about the average depth of the constructed tree.

**Observation 2.** *Theorem 1 is an existential result and the construction outlined above cannot be done in polytime, since it relies on the availability of an optimal multiway tree. However, one can obtain in polytime a tree that is simultaneously an $O(\log n)$-approximation for both worst-case (respectively average) explanation size and depth by replacing the optimal multiway tree by one that approximates within a factor of $O(\log n)$ the worst-case (resp. average) depth, which can be found in polytime (see, e.g., [18] and references therein quoted).*

Although guaranteeing asymptotically the desired optimal approximation, the construction leading to such trees might be wasteful in practice as it involves the use of distinct alphabetic codes to turn each multiway tests into a short sequences of threshold tests. Therefore, we present an alternative approach in Section 5 that achieves the same approximation guarantee and has also very good performance in practice.

## 4.1 Lower bound

Given these positive results, a natural question is whether it is possible to obtain a tree that is optimal for both depth and explanation size. The next result answers this in the negative, and shows that in a way the worst-case bound in Theorem 1 cannot be improved.

**Theorem 2.** *Fix $c$ and $n \geq cst \cdot c$ for a sufficiently large constant $cst$. Then for every $\alpha \in [\frac{1}{2 \log c}, \frac{1}{2}(\frac{\log(n/2)}{\log c} - 1)]$, there is a classification instance with $n$ examples and $c$ classes such that every binary decision tree $T$ for this instance has either*

$$\mathrm{depth}_{wc}(T) > \frac{\alpha}{2} \cdot \mathrm{depth}^*_{wc}$$

*or*

$$\mathrm{expl}_{wc}(T) > \frac{1}{4\alpha} \cdot \log\left(\frac{n}{c}\right) \cdot \mathrm{expl}^*_{wc}.$$

**Remark 1.** *To get a more concrete idea for this lower bound, consider setting $\alpha$ at its upper limit, namely $\alpha \approx \frac{1}{2} \frac{\log n}{\log c}$. In this case we obtain that every tree with $\mathrm{depth}_{wc}(T) \lesssim \frac{1}{4} \frac{\log n}{\log c} \mathrm{depth}^*_{wc}$ must have $\mathrm{expl}_{wc}(T) \geq \Omega((1 - \frac{\log c}{\log n}) \log c) \mathrm{expl}^*_{wc}$, which is $\Omega(\log n) \mathrm{expl}^*_{wc}$ if we set $c = \frac{\log n}{2}$. Comparing this against Theorem 1 and Observation 1 we see an interesting and subtle phenomenon: while you can have a tree with $\mathrm{depth}_{wc}(T) \leq \frac{3 \log n}{\log c} \mathrm{depth}^*_{wc}$ and optimal $\mathrm{expl}_{wc}(T)$, if you require the approximation in the depth to be a constant factor smaller then you must lose a logarithmic factor in the approximation in the explanation size.*

# 5 An efficient and practical algorithm

In this section we design an algorithm, which we name Short Explanaible Rules (SER-DT), that always yields trees of approximately optimal average/worst-case explanation size and depth. Importantly, our method has enough flexibility that it can be tuned to trade-off accuracy and interpretability, as shown in our computational experiments (Section 6).

Similar to other algorithms in the area, ours chooses in each step a split that creates subtrees with "small impurity". However, unlike most such algorithms, it is not completely greedy and allows for the desired extra flexibility in the choices.

To describe the algorithm, consider a set of objects $S \subseteq O$ and let $S(a, i)$ (respectively $S(a, \leq i)$ and $S(a, > i)$) be the set of objects in $S$ with value equal (respectively at most and larger than) $i$ on attribute $a$. Moreover, let $w(S) := \sum_{o \in S} w(o)$ denote its total weight. Let $o, o'$ be a pair of objects in $S$ such that $c(o) \neq c(o')$. We will refer to such a pair as a misclassified pair because if both $o$ and $o'$ reach the same leaf in the decision tree then one of them will be surely misclassified.

We use $P(S)$ to denote the number of misclassified pairs in $S$. This quantity can be thought of as a measure of the amount of work that is needed to reach a correct classification of all objects in $S$ and it has been previously used in [19, 22, 18]. To take into account also the importance/weight of set $S$ we define $\mathtt{wpm}(S) := P(S) \cdot w(S)$ as the weighted pair-wise misclassification of $S$.

As a pre-processing step, before executing SER-DT, each weight $w(o)$ smaller than $w(O)/n^3$ is replaced with $w(O)/(w_{min}n^3)$, where $w_{min}$ is the smallest positive weight among the objects in $O$. This idea (from [35]) is important to guarantee a logarithmic dependence on $n$ instead of $w(O)$. After this preprocessing, SER-DT is called for the set of objects $O$.

The pseudo-code description of SER-DT is presented in Algorithm 1, First SER-DT tries to use **any balanced** test that reduces the weighted pair-wise misclassification of the current set of objects (in the worst case) by at least a $\frac{1}{2}$ factor. In any path of the tree built by SER-DT the amount of these balanced tests is at most logarithmic, so they can be easily handled in our analysis. If no balanced test exists, then the algorithm finds an attribute $a^*$ and value $t^*$ such that in the ternary split $S(a^*, < t)$, $S(a^*, t^*)$, $S(a^*, > t^*)$, only the middle set $S(a^*, t^*)$ has weighted pair-wise misclassification larger than the desired $\frac{1}{2}\mathtt{wpm}(S)$. This 3-way partition is obtained by using two binary splits. Then, the algorithm recurses on each set. A critical issue is to show that in this case some progress is also achieved with the problematic subproblem on $S(a^*, t^*)$ where $\mathtt{wpm}(S(a^*, t^*)) > \frac{1}{2}\mathtt{wpm}(S)$. In fact, the choice of the attribute $a^*$ is such that, the instance $S(a^*, t^*)$ has the minimum weighted pair-wise misclassification among all the attributes and other possible tripartitions. As a result, we can employ a lower bound on the optimum ( Lemma 8 in appendix) that allows us to absorb the cost of the subtree for the subproblem $S(a^*, t)$ in the logarithmic guarantee.

---

**Algorithm 1** SER-DT ($S$ : set of objects)

---

1: **if** all objects in $S$ are assigned to the same class, create a leaf assigned to such class and **return**
2: **if** there is a test $\tau$ that splits $S$ into $S^L$ and $S^R$ such that $\max\{\mathtt{wpm}(S^L), \mathtt{wpm}(S^R)\} \leq \frac{1}{2}\mathtt{wpm}(S)$ **then**
3:     Use any such test, say $\tau = (a, t)$, as the root of the decision tree
4:     Recurse on the children $S(a, \leq t)$ and $S(a, > t)$
5: **else**
6:     Let $a^*$ be an attribute in $\operatorname*{argmin}_a\{\max_i \mathtt{wpm}(S(a, i))\}$
7:     Let $t^*$ be the smallest value of the attribute $a^*$ such that the "left child" $S(a^*, \leq t^*)$ satisfies

$$\mathtt{wpm}(S(a^*, \leq t^*)) \geq \frac{1}{2} \cdot \mathtt{wpm}(S).$$

8:     Use two binary tests to simulate the 3-way split $S(a^*, < t^*)$, $S(a^*, t^*)$, $S(a^*, > t^*)$. More precisely, at the root use a test on attribute $a^*$ that splits $S$ into the sets $S(a^*, < t^*)$ and $S(a^*, \geq t^*)$. Then, apply a test on the right child of the root, currently associated with $S(a^*, \geq t^*)$, creating two new children with objects $S(a^*, t^*)$ and $S(a^*, > t^*)$
9:     Recurse on each of the three leaf nodes in the current tree
10: **end if**

---

The following is the promised guarantee for the average/worst-case depth and explanation size of the trees produced by the algorithm.

**Theorem 3.** *Given an instance $I$, the algorithm SER-DT produces a tree $T$ that satisfies*

1. *(Worst-case metrics)*
   - $\mathrm{depth}_{avg}(T) \leq O\left(\log n\right) \mathrm{depth}^*_{avg}$,
   - $\mathrm{expl}_{avg}(T) \leq O\left(\log n\right) \mathrm{expl}^*_{avg}$.

2. *(Average metrics)*
   - $\mathrm{depth}_{wc}(T) \leq O\left(\log n\right) \mathrm{depth}^*_{wc}$,
   - $\mathrm{expl}_{wc}(T) \leq O\left(\log n\right) \mathrm{expl}^*_{wc}$.

We remark that, in the light of the inapproximability results from [36, 16][2] this theorem says that SER-DT (with the preprocessing step) guarantees the best possible approximation obtainable by a polynomial algorithm with respect to both the measures under consideration: worst/average depth and worst/average explanation size.

## 6  Experiments

In this section, we report the experiments that were carried out to evaluate how our proposed algorithm SER-DT performs in practice.

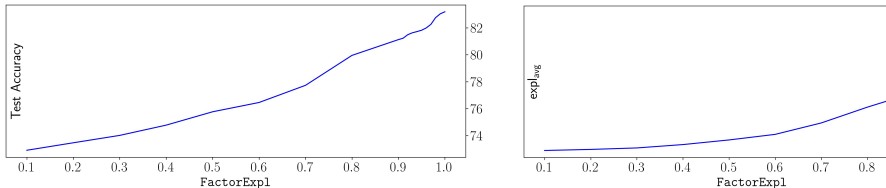

Figure 2: The left (resp. right) image shows the average accuracy (resp. $\mathrm{expl}_{avg}$) over the 20 datasets as a function of `FactorExpl`.

We considered the 20 datasets that appear on Column 1 of Table 1 (see Appendix E for their main characteristics). For all of them, $70\%$ of the examples were used for training and the remaining $30\%$ for testing. Moreover, all the examples (objects) were considered equally important (weight=1/size of training set). During a preprocessing step we converted all categorical attributes into binary attributes via one-hot-encoding. See also Appendix E for more details on the experimental setup.

Recall that in (Line 2) of algorithm SER-DT any test that splits the current set of objects into subsets of small enough `wpm` could be used. We use this flexibility to select a test among these that should further help in obtaining small explanation sizes and high accuracy in practice. To explain our selection, recall the `Gini` impurity measure employed by `CART`. For a set of examples (objects) $S$, each of them labeled with a class in $\{1, \ldots, c\}$, the `Gini` impurity is given by

$$\mathtt{Gini}(S) = 1 - \sum_{i=1}^{c} \left( \frac{|S_i|}{|S|} \right)^2,$$

where $S_i$ is the set of examples of class $i$. Moreover, the weighted `Gini` impurity $\mathtt{Gini}(\tau, \nu)$ induced by a test $\tau$ that divides the set of examples $S$ that reach a node $\nu$ into $S^L$ and $S^R$ is given by

$$\mathtt{Gini}(\tau, \nu) = \frac{|S^L|}{|S|} \mathtt{Gini}(S^L) + \frac{|S^R|}{|S|} \mathtt{Gini}(S^R).$$

Let `FactorExpl` be a hyper-parameter in the range $[0, 1]$. Because we are interested in trees that induce accurate classifiers with short explanation rules, to expand a given leaf $\nu$, in Line 2 of the algorithm we select, among the permissible tests [3] the test $\tau$ that minimizes

$$\mathtt{AdjustedGiniExpl}(\tau, \nu) := \mathtt{I}(\tau, \nu) \times \mathtt{Gini}(\tau, \nu),$$

---

[2]The hardness of approximation holds also for instances with only binary attributes. In this case explanation size coincides with depth and every test can be considered a threshold test.

[3]A permissible $\tau$ must satisfy $\mathtt{Gini}(\tau, \nu) < \mathtt{Gini}$ (Examples that reach $\nu$), in addition to respect the condition of Line 2

where $I(\tau, \nu) = \mathtt{FactorExpl}$ if the attribute associated with test $\tau$ has already appeared in the path from the root to $\nu$, and $I(\tau, \nu) = 1$ otherwise. Since $\mathtt{FactorExpl}$ is used to favour attributes that have already appeared in the path, when $\mathtt{FactorExpl}$ is set to a low value we expect to obtain trees with short explanations but also with lower accuracy, as the effect of the $\mathtt{Gini}$ impurity is reduced.

In terms of stopping rules, we do not expand a leaf $\nu$ if it is either located at depth 6 or if there is no test $\tau$ for which $\mathtt{Gini}(\tau, \nu)$ is smaller than $\mathtt{Gini}(\text{Examples thaat reach } \nu)$. As a post-processing step, whenever two sibling leaves are assigned the same class, we delete them both and leave their parent as a leaf.

In our first experiment we study how the accuracy and the interpretability measures of the trees produced by our algorithm behave when $\mathtt{FactorExpl}$ is varied. Figure 2 shows the average accuracy (left image) and the average explanation size $\mathrm{expl}_{avg}$ (right image) as a function of $\mathtt{FactorExpl}$. More precisely, for the left image, the $y$-value associated with a point $x$ corresponds to the average accuracy on the testing set, calculated over the 20 datasets, when our algorithm is executed with $\mathtt{FactorExpl} = x$. For the right image the same logic holds. As expected, the larger the $\mathtt{FactorExpl}$ the larger the accuracy and the $\mathrm{expl}_{avg}$. The **interesting finding**, however, is that the accuracy increases relatively slow, for $\mathtt{FactorExpl}$ is close to 1, compared with the growth in $\mathrm{expl}_{avg}$ (see the Table 5 in the appendix for experiments with $\mathtt{FactorExpl}$ in the range [0.95,0.99]). This suggests that it is possible to obtain trees that are significantly more interpretable without sacrificing the accuracy.

Table 1: Test Accuracy, $\mathrm{expl}_{avg}$ and $\mathrm{expl}_{wc}$ for $\mathtt{FactorExpl} = 0.97$. Each entry is the average of 10 runs using different seeds to select the examples in the training and testing set. Boldface values indicate a difference of more than 1% (columns 2,3) or a gain of at least 25% in favour of SER-DT (columns 4,5,6 and 7).

| Dataset | Test Accuracy | | $\mathrm{expl}_{avg}$ | | $\mathrm{expl}_{wc}$ | |
|---|---|---|---|---|---|---|
| | SER-DT | CART | SER-DT | CART | SER-DT | CART |
| anuran | 94,8% | 94,7% | 4,78 | 5,24 | 6,0 | 6,0 |
| audit risk | 99,9% | 99,9% | 1,00 | 1,00 | 1,0 | 1,0 |
| avila | 61,5% | **63,2%** | **3,06** | 4,22 | 4,9 | 5,4 |
| banknote | 97,6% | 98,1% | 2,44 | 2,55 | 3,8 | 3,4 |
| bankruptcy polish | 96,6% | 96,9% | **2,56** | 4,63 | 5,6 | 5,9 |
| cardiotocography | 89,5% | 89,8% | 4,30 | 5,30 | 5,9 | 6,0 |
| collins | 13,2% | **15,6%** | **2,13** | 4,76 | **4,4** | 5,9 |
| default credit card | 82,0% | 81,9% | **1,45** | 4,29 | **4,5** | 6,0 |
| dry bean | 90,1% | 89,8% | **3,32** | 4,45 | 5,1 | 6,0 |
| eeg eye state | 74,1% | 73,6% | 3,69 | 4,29 | 5,9 | 6,0 |
| htru2 | 97,7% | 97,7% | **1,20** | 2,03 | 4,3 | 4,9 |
| iris | 94,2% | 93,6% | 1,75 | 1,76 | 3,1 | 3,4 |
| letter recognition | 44,9% | **47,9%** | **3,34** | 5,50 | 5,5 | 6,0 |
| mice | 99,9% | 99,9% | 3,05 | 3,05 | 3,6 | 3,6 |
| obs network | **91,7%** | 89,5% | 3,48 | 4,26 | 5,3 | 5,9 |
| occupancy room | 99,4% | 99,3% | 4,18 | 4,54 | 5,3 | 5,7 |
| online shoppers intention | 89,3% | 89,8% | 3,30 | 4,00 | 5,1 | 6,0 |
| pen digits | **88,6%** | 86,9% | 4,76 | 5,31 | 5,8 | 6,0 |
| poker hand | 52,9% | **55,0%** | **1,80** | 4,30 | **3,8** | 5,1 |
| sensorless | **87,4%** | 80,1% | **2,94** | 4,03 | 4,9 | 5,5 |
| Average | 82,3% | 82,2% | 2,93 | 3,97 | 4,69 | 5,19 |

In our second experiment we compare the results of our method, using $\mathtt{FactorExpl} = 0.97$ with $\mathtt{CART}$ [11]. The value 0.97 is motivated by the above observation. To expand a leaf $\nu$, recall that $\mathtt{CART}$ selects the test $\tau$ for which $\mathtt{Gini}(\tau, \nu)$ is minimum and it only expands $\nu$ if there is a test $\tau$ for which $\mathtt{Gini}(\tau, \nu) < \mathtt{Gini}(\text{Examples that reach } \nu)$. To provide a fair comparison with our algorithm we set the maximum depth to 6 and applied the same aforementioned post-processing. Table 1 shows the accuracy on the testing set as well as the average explanation size $\mathrm{expl}_{avg}$ and the worst-case explanation size $\mathrm{expl}_{wc}$ for all datasets. Each entry in this table is the average of ten runs, where in each of them a different seed is used to split a dataset into training and testing set.

We notice that the accuracy of our method is very close to that obtained by $\mathtt{CART}$, while the gain in terms of the interpretability metrics is significant. On 7 datasets (bold-faced on columns 2 and 3) we

observe a difference larger than $1\%$ on the accuracies; on 3 of them our algorithm outperforms `CART` while on the remaining 4, `CART` is better. In terms of the average explanation size, our algorithm is at least as good as `CART` for all datasets, and for 9 of them (bold-faced on columns 4 and 5) it improves the $\text{expl}_{avg}$ by at least $25\%$. For the $\text{expl}_{wc}$ the gain is also clear. For all datasets, but `Banknote`, our algorithm is at least as good as `CART`. Moreover, for 3 of them (bold-faced on columns 6 and 7), it provides a gain of at least $25\%$. Boxplots for the experiments in Table 1 are provided in Section E.5 of the appendix.

In Appendix E we compare `CART` and SER-DT in terms of $\text{depth}_{avg}$ and $\text{depth}_{wc}$. For $\text{depth}_{avg}$, SER-DT performs better than `CART` while for $\text{depth}_{wc}$ the results are similar. The result for $\text{depth}_{wc}$ is not surprising since we set 6 as the maximum depth in our experiments. Regarding running time, the algorithms present similar behaviour, as it can be verified in Appendix E. This is somehow expected since both consist of mostly a greedy split selection at each node.

In the appendix we also present additional experiments and analyses. In particular, in Section E.8, we evaluate the impact of applying post-pruning to both `CART` and SER-DT. Moreover, in Section E.7, we show comparisons of SER-DT with one of the state of the art decision tree methods that optimize the average depth, namely the EC2 algorithm from [22]: the experiments suggest that SER-DT performs significantly better than EC2 on all metrics.

## 7    Conclusion

In this work, we proposed the explanation size as a new metric to capture intrepretability of decision trees and initiated a principled study of it. We presented upper and lower bound on the trade-off of simultaneously optimizing this new metric and metrics related to depths of the leaves.

We also proposed a practical algorithm that provably approximates the average explanation size and the average depth and showed, via experiments over 20 datasets, that it is competitive with the widely used `CART` algorithms in terms of accuracy while being much better in terms of producing trees with short explanation size.

On the basis of both the theoretical analysis (approximation guarantee) and the performance demonstrated in the empirical studies, we believe that our algorithm (or some variation based on its ideas) can be used to generate accurate and highly interpretable trees for practical applications.

### Acknowledgments and Disclosure of Funding

This study was financed in part by the Coordenação de Aperfeiçoamento de Pessoal de Nível Superior - Brasil (CAPES) - Finance Code 001. The work of the third author is partially supported by CNPq (grant 307572/2017-0), by FAPERJ, grant E- 26/202.823/2018 and by the Air Force Office of Scientific Research under award number FA9550-22-1-0475.

The fourth author is supported by Bolsa de Produtividade em Pesquisa #312751/2021-4 from CNPq, and FAPERJ grant Jovem Cientista do Nosso Estado.

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
