# Appendix

## A Multiway trees and optimal explanation sizes

In this section we prove a characterization of optimal explanation sizes via multiway trees. First recall that the only difference between these and threshold decision trees is that in multiway trees each test corresponds to an attribute $a$, and the answer to the test is the value $a(o)$ that the desired object $o$ has on attribute $a$; each internal node $\nu$ is associated with such a test $\tau_\nu$, and the edges from a node to its children are associated to the possible outcomes of $\tau_\nu$. The worst-case/average depth and explanation size of a multiway tree are defined exactly as for threshold decision trees, see Section 3.

We start by proving this equivalence for the worst-case metrics.

**Lemma 2** (Multiway trees and expl. size, worst-case). *Consider an instance of the classification problem, and let $M^*$ be a multiway tree of minimal worst-case depth for this instance. Then*

$$\text{expl}^*_{wc} = \text{expl}_{wc}(M^*) = \text{depth}_{wc}(M^*).$$

*Proof.* Some quick notation: Given a multiway tree $M$ and one of its nodes $\nu$, let $a_\nu$ be the attribute that is tested on this node. Also let $M_{\nu,i}$ be the subtree of $M$ rooted at the child of $\nu$ reached by objects whose value for attribute $a_\nu$ is $i$. We use $\text{int}(M)$ to denote the set of internal nodes of a tree $M$.

We first show that $\text{expl}^*_{wc} \le \text{expl}_{wc}(M^*) \le \text{depth}_{wc}(M^*)$. For that, we convert $M^*$ into a threshold decision tree that has explanation size at most $\text{depth}_{wc}(M^*)$ by just simulating the multiway tests using multiple threshold tests in the natural way (see Fig.3 for a pictorial example of the following construction). More formally, let $T_{a_\nu}$ be binary search tree over the values of attribute $a_\nu$, that is, it is a threshold decision tree where there is a one-to-one correspondence between its leaves and the values of the attribute $a_\nu$ (i.e. every object $o$ reaches the unique leaf of $T_{a_\nu}$ corresponding to the value $a_\nu(o)$). Then let $T$ be the threshold decision tree obtained from $M^*$ by (starting from the root and proceeding downwards) replacing each internal node $\nu$ of $M^*$ with $T_{a_\nu}$ and identifying the leaf of $T_{a_\nu}$ corresponding to value $a_\nu(\cdot) = i$ with the root of $M^*_{\nu,i}$.

Consider a leaf $\ell_{M^*}$ in $M^*$. Notice that the path $P_{M^*}$ from the root of $M^*$ to $\ell_{M^*}$ induces a path $P_T$ in the tree $T$ that goes from its root to one of its leaves $\ell_T$ (i.e., whenever the path $P_{M^*}$ goes from a node $\nu$ to a subtree $M^*_{\nu,i}$, the path $P_T$ goes from the root of the binary search tree $T_{a_\nu}$ to its leaf corresponding to value $a_\nu(\cdot) = i$). We can see that the set of attributes tested in the paths $P_{M^*}$ and $P_T$ is exactly the same, and thus the leaves $\ell_{M^*}$ and $\ell_T$ have the same explanation size. Looking at the largest explanation size of the leaves in these trees, we get that $\text{expl}_{wc}(T) = \text{expl}_{wc}(M^*)$. This gives

$$\text{expl}^*_{wc} \le \text{expl}_{wc}(T) = \text{expl}_{wc}(M^*) \le \text{depth}_{wc}(M^*) \tag{1}$$

as desired.

Now we show that $\text{depth}_{wc}(M^*) \le \text{expl}^*_{wc}$, which together with (1) above gives the desired equalities in the lemma. Let $T$ be a threshold decision tree achieving the optimal explanation size $\text{expl}^*_{wc}$. We will show how to build a multiway decision tree $M$ such that $\text{depth}_{wc}(M) = \text{expl}_{wc}(T) = \text{expl}^*_{wc}$ (refer to Fig. 4 for an example of the construction).

We first recursively turn each test of $T$ into a multiway test on the same attribute in the natural way. More precisely, this can be done in a recursive way: If $T$ is a leaf, then define the multiway tree $\tilde{M} = T$. Otherwise, let $(a, t)$ be the test at root of $T$, and let $T_<$ and $T_\ge$ be the subtrees of $T$ rooted at the children of the root and associated with the two results of the root-test, i.e., $T_<$ (resp. $T_\ge$) is the subtree reached by the objects $o$ such that $a(o) < t$ (resp. $a(o) \ge t$). Then $\tilde{M}$ is obtained by putting the multiway test on attribute $a$ on its root $r$, and for each value $i$ of the attribute $a$, if $i < t$ (resp. $i \ge t$) the $i$-th child of the root $r$ is the subtree obtained by recursively applying this construction to $T_<$ (resp. $T_\ge$).

Since this construction simply converts each threshold test $(a, t)$ in $T$ into a multiway test on $a$, it is easy to see that

$$\text{expl}_{wc}(\tilde{M}) = \text{expl}_{wc}(T) = \text{expl}^*_{wc}. \tag{2}$$

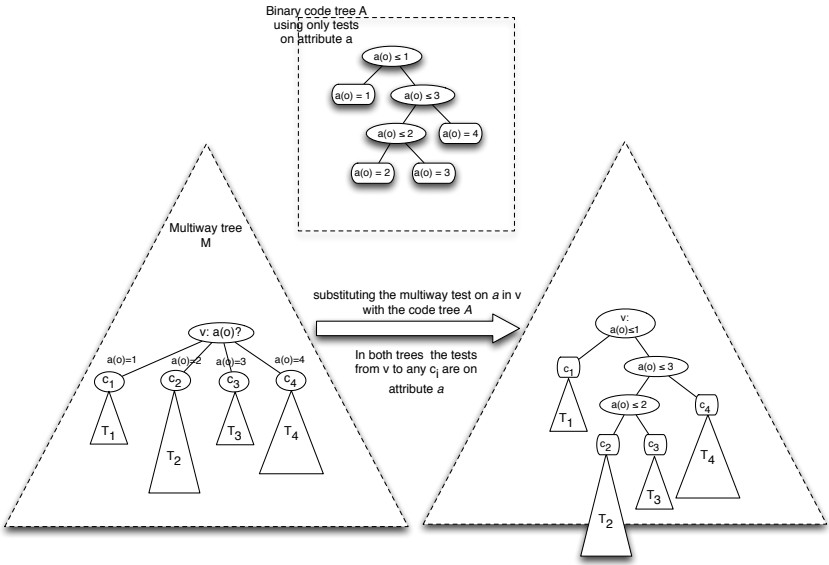

Figure 3: From multiway tree to threshold tree

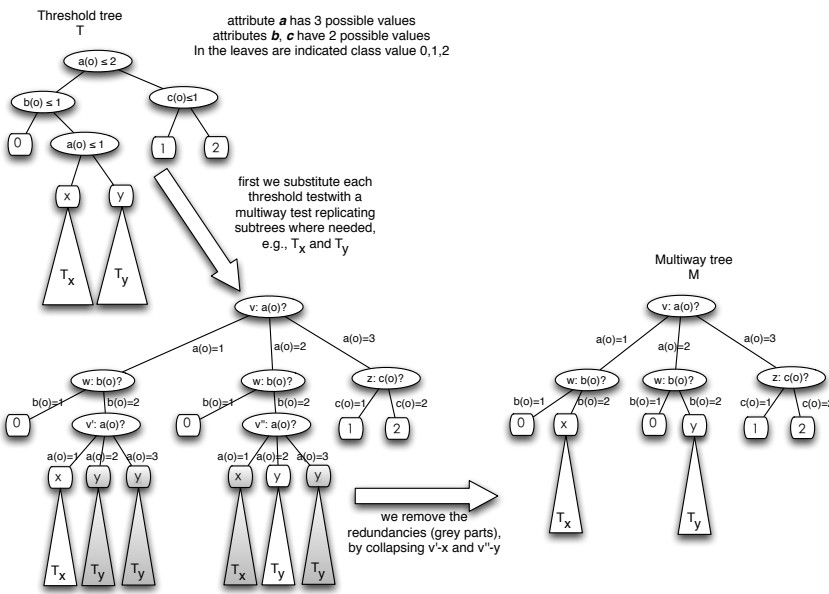

Figure 4: From threshold tree to multiway tree

However, the multiway tree $\tilde{M}$ may test the same attribute multiple times on a root-to-leaf path, because $T$ may do so, and so we can have $\mathrm{depth}_{wc}(\tilde{M}) > \mathrm{expl}_{wc}(\tilde{M})$. But in $\tilde{M}$ these multiple tests are redundant, because a single multiway test reveals complete information about the attribute.

So we now remove these redundancies to obtain a multiway tree $M$ with $\mathrm{depth}_{wc}(M) = \mathrm{expl}_{wc}(\tilde{M})$, as follows: If there is a node $\nu \in \mathrm{int}(\tilde{M})$ and a descendant $\nu'$ in one of the child-subtree $T_{\nu,i}$ that tests the same attribute as $\nu$ (i.e., $a_{\nu'} = a_\nu$), we replace the subtree $T_{\nu'}$ with the subtree $T_{\nu',i}$. Notice that since $\nu' \in T_{\nu,i}$, any object reaching node $\nu'$ has $a(\cdot) = i$, and thus would move next to the subtree $T_{\nu',i}$; our replacement operation just bypassed the node $\nu'$, and so we can see that the multiway tree obtained remains equivalent to $\tilde{M}$ for the classification problem. Repeat this replacement operation

until no root-to-leaf path contains two nodes that test on the same attribute. Let $M$ be the multiway tree obtained at the end.

Since no attribute is repeatedly tested in such paths, for every leaf $\ell \in \text{leaf}(M)$ we have $\text{depth}(\ell) = \text{expl}(\ell)$, and hence

$$\text{depth}_{wc}(M) = \text{expl}_{wc}(M) \leq \text{expl}_{wc}(\tilde{M}),$$

the inequality following because $M$ was obtained by deleting nodes of $\tilde{M}$. Together with inequality (2) and the fact that $M^*$ is depth-optimal, this gives

$$\text{depth}_{wc}(M^*) \leq \text{depth}_{wc}(M) \leq \text{expl}^*_{wc},$$

as desired. This concludes the proof of the lemma. $\qquad\qquad\qquad\qquad\qquad\qquad\square$

The same argument also shows the following analogous result about the average depth and explanation size.

**Lemma 3** (Multiway trees and expl. size, average)**.** *Consider an instance of the classification problem, and let $M^*$ be a multiway tree of minimal average depth for this instance. Then*

$$\text{expl}^*_{avg} = \text{expl}_{avg}(M^*) = \text{depth}_{avg}(M^*).$$

# B   Proof of Theorem 1

We prove the guarantees for the worst-case and average metrics (Items 1 and 2 of the theorem) separately. As before, given a tree $T$ and one of its nodes $\nu$, we use $T_\nu$ to denote the subtree rooted at $\nu$. We also use $n(\nu)$ to denote the number of objects of the instance that reach node $\nu$ (and hence some leaf in the subtree rooted at $\nu$); sometimes we also use a subscript $n_T(\nu)$ to make it clearer which tree we are talking about in order to avoid any confusion.

## B.1   Proof of Item 1: Worst-case metrics

Recall that in this item we want a threshold tree $T$ satisfying:

$$\text{expl}_{wc}(T) = \text{expl}^*_{wc} \tag{3}$$
$$\text{depth}_{wc}(T) \leq 2 \, \text{depth}^*_{wc} + \log n. \tag{4}$$

Using Lemma 2, let $M^*$ be an optimal multiway tree with respect to both worst-case explanation size and depth, and hence satisfying $\text{depth}_{wc}(M^*) = \text{expl}_{wc}(M^*) = \text{expl}^*_{wc}$. We show how to transform $M^*$ into a binary tree $T$ with $\text{expl}_{wc}(T) = \text{expl}_{wc}(M^*)$ and $\text{depth}_{wc}(T) \leq 2 \, \text{depth}_{wc}(M^*) + \log n$ (which then will give (3)-(4)). Consider an internal node $\nu$ of $M^*$, which tests an attribute $a$ and has as children $ch_1, ch_2, \ldots$. Let $d_i := \lceil \log \frac{n(\nu)}{n(ch_i)} \rceil + 1$ and notice that

$$\sum_i 2^{-(\lceil \log \frac{n(\nu)}{n(ch_i)} \rceil + 1)} \leq \frac{1}{2} \sum_i 2^{-\log \frac{n(\nu)}{n(ch_i)}} = \frac{1}{2}, \tag{5}$$

the last equation following because $\sum_i n(ch_i) = n(\nu)$. Then applying the result about the existence of alphabetic codes (Lemma 1) with the attribute $a$ and considering all objects that reach the subtree $M^*_{ch_i}$ as a single object, we can get a threshold decision tree $A$ with one leaf for each $ch_i$ such that: 1) The leaf corresponding to $ch_i$ is at height (in $A$) $d_i$, and; 2) Every object that in $M^*$ reaches node $ch_i$, in $A$ reaches the leaf corresponding to $ch_i$. It will be convenient to call $A$ a code tree. Then, we proceed like in the proof of the first part of Lemma 2 (see also Fig. 3): replace the node $\nu$ of $M^*$ by the tree $A$ (with the root of the subtree $M^*_{ch_i}$ identified with the leaf of $A$ corresponding to $ch_i$). Perform this replacement for every node of $M^*$, and let $T$ be the resulting threshold tree.

We refer to the nodes of $T$ that are the roots of the code trees $A$ used in the process as the original nodes. The motivation for this terminology is that these nodes correspond to the nodes in the original tree $M^*$. Observe that if $v$ is an original node in $T$ and $u$ its corresponding node in $M^*$, then $n_T(v) = n_{M^*}(u)$.

We first claim that $\text{expl}_{wc}(T) = \text{expl}_{wc}(M^*)$ (and hence, by definition of $M^*$, also equals $\text{depth}^*_{wc}$). Let $\ell_T$ be a leaf of $T$ and $\ell_{M^*}$ be the corresponding leaf of $M^*$. By construction, the set of attributes

tested on the path reaching $\ell_T$ is the same as the set of attributes tested on the path reaching $\ell_{M^*}$, since the tests associated to the nodes $\nu$ on the latter path have been expanded in $T$ to a path in the code tree used to replace $\nu$ and such a code tree uses only tests on the same attribute tested at $\nu$. Therefore, we have that the explanation size of $\ell_T$ is the same as that of $\ell_{M^*}$. Considering the leaves with largest explanation sizes in these trees, we get $\mathrm{expl}_{wc}(T) = \mathrm{expl}_{wc}(M^*) = \mathrm{expl}_{wc}^*$ as we wanted. This gives the desired equality (3).

Now we claim that $\mathrm{depth}_{wc}(T) \leq 2\,\mathrm{depth}_{wc}(M^*) + \log n$. For that, consider any root-to-leaf path $P$ in $T$, and let $\nu_1, \ldots, \nu_p$ be the underline{original nodes} in this path. Since both the root and leaves of $T$ are underline{original nodes}, we see that $\nu_1$ and $\nu_p$ are the first and last nodes of the path $P$, respectively. Letting $d(u, v)$ denote the distance (in number of edges) between two nodes $u, v$ in $T$, we see that the length (number of internal nodes, or equivalently, edges) of $P$ can be written as

$$|P| = \sum_{i=1}^{p-1} d(\nu_i, \nu_{i+1}). \tag{6}$$

Moreover, letting $A$ be the alphabetic code tree used in the construction of $T$ that is rooted at $\nu_i$, we have that the subpath between the underline{original nodes} $\nu_i$ and $\nu_{i+1}$ coincides with the path of $A$ between its root and its leaf corresponding to $\nu_{i+1}$. Then, by the construction of $A$ we have

$$d(\nu_i, \nu_{i+1}) \leq \left\lceil \log \frac{n(\nu_i)}{n(\nu_{i+1})} \right\rceil + 1.$$

Replacing this bound in (6) gives

$$|P| \leq \sum_{i=1}^{p-1} \left( \left\lceil \log \frac{n(\nu_i)}{n(\nu_{i+1})} \right\rceil + 1 \right) \tag{7}$$

$$\leq 2(p-1) + \sum_{i=1}^{p-1} \log \frac{n(\nu_i)}{n(\nu_{i+1})} \;\leq\; 2(p-1) + \log n$$

$$\leq 2\,\mathrm{depth}_{wc}(M^*) + \log n, \tag{8}$$

where the last inequality follows because all of the underline{original nodes} $\nu_1, \ldots, \nu_{p-1}$ are (or correspond to) internal nodes in a root-to-leaf path in $M^*$, and so $\mathrm{depth}_{wc}(M^*) \geq p - 1$. Since the bound (8) holds for every root-to-leaf path $P$ of $T$, we obtain that $\mathrm{depth}_{wc}(T) \leq 2\,\mathrm{depth}_{wc}(M^*) + \log n$, proving the claim. Since by definition of $M^*$ we have $\mathrm{depth}_{wc}(M^*) = \mathrm{expl}_{wc}^*$, which is at most $\mathrm{depth}_{wc}^*$, this proves the desired inequality (4).

This concludes the proof of Item 1 of Theorem 1.

### B.2  Proof of Item 2: Average metrics

We now prove Item 2 of Theorem 1, regarding the underline{average} explanation size and depth, instead of the worst-case ones, namely we want a threshold tree $\overline{T}$ satisfying:

$$\mathrm{expl}_{avg}(T) \;=\; \mathrm{expl}_{avg}^* \tag{9}$$

$$\mathrm{depth}_{avg}(T) \;\leq\; 2\,\mathrm{depth}_{avg}^* + W \log n, \tag{10}$$

where $W = \sum_o w(o)$ denote the total weight of all objects. The proof is almost identical as that of the previous section, the only difference is that the weights of the items will now be taken into account when defining the $d_i$'s in the code trees, and we use entropy-based calculations to argue about the average depth of the constructed tree.

Given a tree $T$ and one of its nodes $\nu$, let $w(\nu)$ denotes the total weights of the objects reaching node $\nu$ (we add the subscript $w_T(\nu)$ when we want to emphasize which tree we are referring to).

Let $M^*$ be the multiway tree given by Lemma 3, which satisfies the guarantees $\mathrm{depth}_{avg}(M^*) = \mathrm{expl}_{avg}(M^*) = \mathrm{expl}_{avg}^*$ for the average metrics. We first note that one can compute the average cost of a tree by summing up the weights of its internal nodes, for example

$$\mathrm{depth}_{avg}(M^*) = \sum_{\nu \in \mathrm{int}(M^*)} w(\nu), \tag{11}$$

where again we use $\text{int}(\cdot)$ to denote the set of internal nodes of a tree.

We now show how to transform $M^*$ into a threshold decision tree $T$ of small average explanation size and depth. For an internal node $\nu$ of $M^*$, which tests an attribute $a_\nu$ and has as children the nodes $\nu_1, \nu_2, \ldots$, let $p^\nu$ be the probability distribution $\left(\frac{w(\nu_1)}{w(\nu)}, \frac{w(\nu_2)}{w(\nu)}, \ldots, \right)$ induced by the partition at $\nu$. Then consider the code tree $T_\nu$ given by Lemma 1 with the attribute $a$ and considering all objects that reach the subtree $M^*_{\nu_i}$ as a single object, but such that the leaf in $T_\nu$ corresponding to $\nu_i$ is now at depth (in $T_\nu$) $d_i := \lceil \log \frac{w(\nu)}{w(\nu_i)} \rceil + 1$ (as in (5), we can see that such $d_i$'s satisfy the requirement of the lemma). Then replace the node $\nu$ of $M^*$ by the tree $T_\nu$ (with the root of the subtree $M^*_{\nu_i}$ identified with the leaf of $T_\nu$ corresponding to $\nu_i$). Perform this replacement for every node of $M^*$, and let $T$ be the resulting threshold tree.

Again we refer to the nodes of $T$ that are the roots of the code trees used in the process as the original nodes, and observe that if $v$ is an original node in $T$ and $u$ its corresponding node in $M^*$, then $w_T(v) = w_{M^*}(u)$.

Just as before, the set of attributes tested on the path in $T$ from its root to one of its leaves $\ell_T$ is the same as the set of attributes tested on the path in $M^*$ from its root to the leaf $\ell_{M^*}$ that corresponds to $\ell_T$. This implies that $\text{expl}_{avg}(T) = \text{expl}_{avg}(M^*)$, which by definition of $M^*$ also equals $\text{expl}^*_{avg}$. This proves inequality (9).

To prove inequality (10), let $\text{leaf}(M^*) = \{\ell_1, \ldots, \ell_m\}$ be the set of leaves of the multiway tree $M^*$. Let $p^L := (\frac{w(\ell_1)}{W}, \ldots, \frac{w(\ell_m)}{W})$ be the probability distribution induced on these leaves. Also, let $p(\nu) := \frac{w(\nu)}{W}$ be the sum of the probabilities of the leaves in the subtree rooted at $\nu$. Given a distribution $p$, we use $\text{H}(p) := \sum_i p_i \log \frac{1}{p_i}$ to denote its Shannon entropy. The following fact records a well-known property that allows to compute the Shannon entropy of the leaf distribution as the weighted average of the entropy of the node distributions (see, e.g., [25, Chapter 3]).

**Fact 1.** *It holds that* $\text{H}(p^L) = \displaystyle\sum_{\nu \in int(M^*)} p(\nu) \cdot \text{H}(p^\nu).$

Now, we can compute the average depth of the tree $T$ as follows: Since in $T$, each node $\nu$ of $M^*$ is replaced by the code tree $T_\nu$, the corresponding contribution to the average depth of $T$ is given by the average depth of $T_\nu$, namely

$$\text{depth}_{avg}(T_\nu) = \sum_i w(\nu_i) \cdot d_i$$

$$= \sum_i w(\nu_i)\left(\left\lceil \log \frac{w(\nu)}{w(\nu_i)} \right\rceil + 1\right) \le \sum_i w(\nu_i) \log \frac{w(\nu)}{w(\nu_i)} + 2w(\nu). \quad (12)$$

We can use this in the computation of the average depth of the tree $T$, decomposing it into the contributions of the code trees that have been used to replace the nodes of $M^*$:

$$\text{depth}_{avg}(T) = \sum_{\nu \in int(T)} w(\nu) = \sum_{\nu \in int(M^*)} \sum_{u \in int(T_\nu)} w(u) \quad (13)$$

$$= \sum_{\nu \in int(M^*)} \text{depth}_{avg}(T_\nu)$$

$$\le \sum_{\nu \in int(M^*)} w(\nu) \sum_i \frac{w(\nu_i)}{w(\nu)} \log \frac{w(\nu)}{w(\nu_i)}$$

$$+ 2 \sum_{\nu \in int(M^*)} w(\nu) \quad (14)$$

$$= W \sum_{\nu \in int(M^*)} p(\nu) \cdot \text{H}(p^\nu) + 2\,\text{depth}_{avg}(M^*) \quad (15)$$

$$= W \cdot \text{H}(p^L) + 2\,\text{depth}_{avg}(M^*) \quad (16)$$

$$\le W \log |\text{leaf}(M^*)| + 2\,\text{depth}_{avg}(M^*) \quad (17)$$

$$\le W \log n + 2\,\text{depth}^*_{avg},$$

where the first line follows from (11); inequality (14) follows from using (12) to upper bound $\text{depth}_{avg}(T_\nu)$ and then multiplying and dividing the first summation by $w(\nu)$; equation (15) follows again from (11); (16) follows from Fact 1; finally, inequality (17) follows from the standard entropy rank bound (i.e., the entropy of a distribution over $m$ objects is at most $\log m$). This proves inequality (10), and concludes the proof of Item 2 of Theorem 1.

## C  Proof of Theorem 2

**Construction of the hard instance.**    We will construct a deterministic instance for the lower bound as follows. There are $n$ objects, and their classes are from the set $C = \{0, 1, \ldots, c - 1\}$ and "alternate": object 1 has class 0, object 2 class 1, object 3 class 2, ..., object $c$ class $c - 1$; then this repeats, object $c + 1$ has class 0, etc. We consider throughout that $n$ is at least $cst \cdot c$ for a sufficiently large constant $cst$.

The first attribute in the instance, named $A$, has $n$ values and gives the identity of the objects (i.e., object $i$ has $A$-value equal to $i$).

The definition of the other attributes is more intricate and we first give their intuition and motivation. First, there is a parameter $t \in (\frac{2c}{n}, \frac{1}{2})$ (these bounds are important so that we can ignore the floor in $\lfloor t\frac{n}{c} \rfloor$). There are several binary attributes each of which discriminates a specific $(1 - t)$-fraction of the objects of one of the classes $\neq 0$ from all the other objects (a more detailed description is given below). The idea is that the top levels of a decision tree of optimal height (which will only use tests on binary attributes) will have the following effect:

1. All of objects of class 0 will traverse the leftmost path

2. Only $t\frac{n}{c}$ of the objects of each class $\neq 0$ will follow this left path

3. All the other $(1 - t)\frac{n}{c}$ objects of each class $\neq 0$ are classified correctly by the top levels of the tree.

Thus, the top of the tree "peels off" (namely, it completes the classification of) a $(1 - t)$ fraction of the objects of each class $\neq 0$, and what is left to be "solved" is on the leftmost path. Then, starting from the last node on the leftmost path of these top levels the tree again will use a similar subtree to peel off another $(1 - t)$ fraction of the objects of each class $\neq 0$, etc. The key to the lower bound is that for any decision tree not employing all these tests on the distinct binary attributes (in the attempt of reducing the number of the tests on binary attributes, and equivalently, the explanation size), the objects (of class $\neq 0$) that such not-performed tests would have correctly classified, will end up on the leftmost path together with the object of class 0. But then tests on the $A$ attribute will need to be used to separate these objects and classify them correctly. Crucially, the objects separated by one of these binary attribute tests will be "uniformly spread out" over $[n]$, so they will "alternate" with the objects of class 0; this means that you need a lot of threshold tests on the attribute $A$ to separate these remaining 0-class and non-0-class objects, paying a lot in terms height.

To make this more precise, let us define a round of a decision tree as the sequence of levels that are meant to achieve the result described in the three-item list above. The sets of objects $S_i \subseteq [n]$ that will be peeled off at round $i$ need to have the following "spread out" property guaranteed by the next lemma, whose proof is deferred to Appendix C.1).

**Lemma 4.** *There are sets of objects $S_1, \ldots, S_w$ (with $w = \frac{\log(n/c)}{\log(1/t)}$) such that:*

- *These sets partition the set of elements of class $\neq 0$*

- *(Spread out) For each $i$, class $\chi \neq 0$, and interval $U \in [n]$, the set $S_i \cap U$ has at least*

$$\frac{1}{2}\frac{|U|}{n}t^{i-1}(1 - t)\frac{n}{c} - 13\log(n/c)$$

*objects of class $\chi$.*

Given these "peeling-off" sets, we can finally conclude the definition of the remaining attributes in the instance: in addition to attribute $A$, there is a binary attribute $(i, j)$ for each $i \in [\log(n/c)/\log(1/t)]$ and $j \in [\log c]$ such that the test $(i, j)$

- sends to the right all the objects in $S_i$ whose class have $j$-th bit equal to 1
- sends to the left all other objects (in particular all those outside of $S_i$).

This concludes the description of the instance.

**Analysis.** Given an integer $x$, we use $bits(x)$ to denote the binary expansion of $x$ (as a vector). Given a (partial) decision tree, when all objects that reach a given leaf have the same class we say that the tree classifies them correctly (since this can be naturally obtained by assigning their class to the leaf).

We start by showing that this instance has a tree of "short" height.

**Lemma 5.** $\mathrm{depth}^*_{wc} \leq \frac{\log(n/c)\log c}{\log(1/t)}$.

*Proof.* We construct such a short tree that satisfies the bound, hence a fortiori it must also hold for the optimum. Again, we use the term round to refer to a group of consecutive levels, where for each level there is a specific test which is performed in all nodes of that level. (Round 1) Build a complete binary tree of depth $\log c$, where at each node on level $j$ the test $(1, j)$ is performed. We claim that these tests peel off the set $S_1$, and classify all of these objects correctly. To see this, observe that these tests give a tree with $c$ leaves, each associated with a vector $y \in \{0, 1\}^{\log c}$ corresponding to the outcome of these tests (where 0 means "go left" and 1 means "go right"). The main observation is that every object of $S_1$ of class $\chi \in \{1, \ldots, (\log c) - 1\}$ will traverse to the leaf corresponding to vector $y = bits(\chi)$. In addition, all objects outside of $S_1$ will end up in the left-most leaf (i.e. the one with vector $y = (0, \ldots, 0)$). With this we can conclude that:

1. All leaves except the left-most one have only objects with the same class (so we do not need to continue splitting these leaves). These leaves contain all elements in $S_1$ (recall that $S_1$ does not have any element of class 0).

2. The leftmost leaf has exactly all elements outside of $S_1$.

(Round 2) We need to further split this leftmost leaf. For that, build a new complete binary tree of depth $\log c$, where at each node on level $j$ test $(2, j)$ is performed. Again by the same argument we get another tree with $c$ leaves, where all objects in $S_2$ will be peeled out and only the elements in $[n] - (S_1 \cup S_2)$ are remaining on the leftmost leaf.

Keep repeating these rounds—each round $i = 1, \ldots, w$, corresponding to executing tests $\{(i, j)\}_{j=1,\ldots,\log c}$—until you finish peeling off all sets $S_i$. At this point the leftmost path has only objects $[n] - (S_1 \cup S_2 \cup \ldots \cup S_w)$, which are all objects of class 0, so they are correctly classified by this leaf. Also, by construction all the remaining objects $S_1 \cup S_2 \cup \ldots \cup S_w$ are also classified correctly in the rest of the tree. So we have obtained a valid tree of total height $w \cdot \log c = \frac{\log(n/c)\log c}{\log(1/t)}$, as desired. $\square$

Now we give a lower bound on the depth of any tree that solves the problem.

**Lemma 6.** *Consider a tree that solves the instance described above. Then there is a root-to-leaf path $P$ such that*

$$K_A + K_B \frac{\log(1/t)}{\log c} \geq \frac{1}{2} \log \frac{n}{c}. \tag{18}$$

*where $K_A$ and $K_B$ are the number of A tests and binary tests (respectively) done on this path.*

*Proof.* We construct the path $P$ as follows: Start at the root of the tree; if the current node $\nu$ corresponds to a test on a binary attribute, go to the left; if it corresponds to a threshold test on the attribute $A$, go to the side where most examples would go if we ignore the tests on the **binary attributes** performed before node $\nu$ (but we do take into account the previous tests on attribute $A$).

We claim that unless (18) holds, at least one object of class 0 and one object of another class $\neq 0$ would reach the end of the path $P$, contradicting that the tree correctly classifies each object.

To see this claim, first consider executing all the binary tests of $P$ and let us see which objects survive (i.e., the result of each test on $P$ make them continue to the next node on $P$). First, all objects of class 0 survive these tests (since $P$ only turns left on binary tests). Moreover, there are $\frac{\log(n/c)\log c}{\log 1/t}$ binary tests $(i,j)$ and $P$ is only doing $K_B$ of them. Thus,

$$\frac{\log(n/c)\log c}{\log(1/t)} - K_B$$

tests $(i,j)$ are not done. Since there are $\log c$ of these $j$'s, an averaging argument shows that there is one $j$ where at least

$$\frac{1}{\log c}\left[\frac{\log(n/c)\log c}{\log(1/t)} - K_B\right] = \frac{\log(n/c)}{\log(1/t)} - \frac{K_B}{\log c}$$

tests $\{(i,j)\}_i$ are not done; let us denote it with $j^*$. Since the $i$'s range from 1 to $\frac{\log(n/c)}{\log(1/t)}$, this implies that there is one test $(i,j^*)$ that is not performed for some $i \leq \frac{K_B}{\log c}$. Denote this test by $(i^*, j^*)$.

Look at the objects in the set $S_{i^*}$ with class $\chi$ where $bits(\chi) = (0, \ldots, 0, 1, 0, \ldots, 0)$ (with 1 in the $j^*$-th position). Call this set $S_{i^*}^\chi$. Since all binary tests other than $(i^*, j^*)$ send the objects $S_{i^*}^\chi$ to the left, again we see that all of these objects survive the binary tests of $P$. In summary: if we just perform the binary tests of $P$, all $\frac{n}{c}$ objects of class 0 survive (call them $S^0$) as well as all objects of the set $S_{i^*}^\chi$.

Now we execute all the threshold tests on the attribute $A$ that are performed along the path $P$. The result of these tests is to select/isolate the objects in some interval $U$ of $[n]$. That is, the objects $(S^0 \cup S_{i^*}^\chi) \cap U$ survive **all** the tests in the path $P$.

By construction of $P$, each one of such tests on the attribute $A$ at most halves the size of the interval of objects selected by the previous tests on $A$, hence $U$ has size at least $\frac{n}{2^{K_A}}$. Thus, since one every $k$ object has class 0, it is immediate to see that

$$|S^0 \cap U| \geq \frac{|U|}{k} - 2 \geq \frac{1}{2^{K_A}}\frac{n}{c} - 2, \tag{19}$$

that is, at least these many objects of class 0 survive. Moreover, by Lemma 4 the number of surviving objects of class $\chi$ is at least:

$$\begin{aligned}
|S_{i^*}^\chi \cap U| &\geq \frac{1}{2}\frac{|U|}{n}t^{i^*-1}(1-t)\frac{n}{c} - 13\log(n/c) \\
&\geq \frac{1}{2}\frac{|U|}{n}\cdot t^{i^*}\cdot\frac{n}{c} - 13\log(n/c) \\
&\geq \frac{1}{2^{K_A+1}}\cdot t^{\frac{K_B}{\log c}}\cdot\frac{n}{c} - 13\log(n/c). \tag{20}
\end{aligned}$$

By assumption that the decision tree correctly classifies each object, we cannot have both classes surviving the path $P$. Hence, at least one of the RHSs in (19) or (20) has to be less than 1. By observing that the RHS of (20) is not larger than the RHS of (19), it must hold that

$$\frac{1}{2^{K_A+1}}\cdot t^{\frac{K_B}{\log c}}\cdot\frac{n}{c} - 13\log(n/c) \leq 1$$

so in particular (since $n \geq cst\cdot c$ with $cst$ a large constant)

$$\frac{1}{2^{K_A+1}}\cdot t^{\frac{K_B}{\log c}}\cdot\frac{n}{c}$$

$$\begin{aligned}
&\leq 14\log(n/c) \stackrel{\log}{\equiv} (K_A+1) + K_B\cdot\frac{\log(1/t)}{\log c} \\
&\geq \log(n/c) - \log(14\log(n/c)).
\end{aligned}$$

Again using the fact that $n \geq cst\cdot c$ for a sufficiently large constant, of the this implies the cleaner bound

$$K_A + K_B\cdot\frac{\log(1/t)}{\log c} \geq \frac{1}{2}\log(n/c),$$

which concludes the proof. $\square$

Now we are ready to prove the main theorem.

*Proof of Theorem 2.* Set $t$ such that $\alpha = \frac{1}{2}\frac{\log 1/t}{\log c}$, i.e., $t = \frac{1}{c^{2\alpha}}$. Since we assumed $\alpha \in [\frac{1}{2\log c}, \frac{1}{2}(\frac{\log(n/2)}{\log c} - 1)]$ we have that $t \in [\frac{2c}{n}, \frac{1}{2}]$, as required by the above argument. Then consider the instance $\mathcal{I}$ defined above for this value of $t$.

Consider any binary tree $T$ solving the instance $\mathcal{I}$ that is an $\frac{\alpha}{2}$-approx for the depth. It suffices to show that this tree has $\mathrm{expl}_{wc}(T) \geq \frac{1}{4\alpha}\log(n/c)\mathrm{expl}^*_{wc}$, which means $\mathrm{expl}_{wc}(T) \geq \frac{1}{4\alpha}\log(n/c)$ (since $\mathrm{expl}^*_{wc} = 1$ for this instance, by using only tests on attribute $A$).

For that, look at the path of $T$ given by Lemma 6. Since $T$ is an $\frac{\alpha}{2}$-approx for the height, we have in particular (using Lemma 5)

$$K_A \leq \frac{\alpha}{2}\mathrm{depth}^*_{wc} \leq \frac{1}{4}\frac{\log 1/t}{\log c} \cdot \frac{\log(n/c)\log c}{\log 1/t} = \frac{1}{4}\log(n/c).$$

But then from Lemma 6 we have

$$K_B\frac{\log 1/t}{\log c} \geq \frac{1}{2}\log\frac{n}{c} - K_A, \text{ i.e., } K_B \geq \frac{1}{4\alpha}\log\frac{n}{c}.$$

Since each binary test in $K_B$ contributes one unit to the explanation size, this proves that the explanation size of $T$ is at least $\frac{1}{4\alpha}\log(n/c)$ as desired. $\qquad\square$

### C.1 Proof of Lemma 4

The lemma will be a direct consequence of the following result.

**Lemma 7.** *Consider $[m]$ and positive scalars $m_i$ such that $m_1 + \ldots + m_w \leq m$, with $w \leq m$. Then there are sets $V_1, \ldots, V_w \subseteq [m]$ such that:*

1. *The $V_i$'s partition $[m]$*

2. *For every interval $J \subseteq [m]$ and $i$, we have*

$$|V_i \cap J| \geq \frac{1}{2}\frac{|J|}{m} \cdot m_i - 12\log m.$$

*Proof.* Without loss of generality assume that $\sum_i m_i = m$, by increasing one of the $m_i$'s if necessary. Construct the sets $V_i$'s randomly as follows: independently for each $j \in [m]$, randomly put $j$ in one of the sets $V_i$ so that $\Pr(j \in V_i) = \frac{m_i}{m}$. Then by definition the $V_i$'s so created partition $[m]$, giving Item 1 of the lemma.

For Item 2, take any interval $J$. First note that

$$\mathbb{E}|V_i \cap J| = \sum_{j \in J}\Pr(j \in V_i) = \frac{|J|}{m} \cdot m_i.$$

Moreover, using the multiplicative Chernoff bound (see, e.g., [43, Chapter 4]) we have that for a fixed $i$,

$$\Pr\left(|V_i \cap J| \leq \frac{1}{2}\mathbb{E}|V_i \cap J|\right) \leq e^{-\frac{\mathbb{E}|V_i \cap J|}{8}} = \exp\left(-\frac{1}{8}\frac{|J|}{m} \cdot m_i\right).$$

Moreover, for any $\delta \in (0,1)$, by considering the cases $\frac{1}{8}\frac{|J|}{m} \cdot m_i \geq \log(1/\delta)$ and $\frac{1}{8}\frac{|J|}{m} \cdot m_i \leq \log(1/\delta)$ we see that this implies

$$\Pr\left(|V_i \cap J| \leq \frac{1}{2}\left[\mathbb{E}|V_i \cap J| - 8\log(1/\delta)\right]\right) \leq \delta.$$

Taking a union bound over all $i \in [w]$ (recall $w \leq m$) and over all intervals $J$ (there are less than $m^2$ of them) we get that with probability strictly more than $1 - m^3\delta$ we have

$$|V_i \cap J| > \frac{1}{2}\left[\mathbb{E}|V_i \cap J| - 8\log(1/\delta)\right] \qquad \forall i, \forall \text{ intervals } J \subseteq [m].$$

Setting $\delta = \frac{1}{m^3}$ shows that there is a scenario where

$$|V_i \cap J| > \frac{1}{2}\frac{|J|}{m} \cdot m_i - 12\log m \qquad \forall i, \forall \text{ intervals } J \subseteq [m].$$

This proves the lemma. $\qquad\square$

*Proof of Lemma 4.* Bucket the objects $[n]$ in buckets $B_1, \ldots, B_{n/c}$, each of size $c$ (with consecutive objects), that is, we have $B_1 = \{1, 2, \ldots, c\}$, $B_2 = \{c + 1, \ldots, 2c\}$, ... until bucket $B_{n/c}$ (for the sake of simplifying the notation, we assume that $n/c$ is an integer). Set $w = \frac{\log(n/c)}{\log(1/t)}$ and apply the previous lemma with parameter $m_i = t^{i-1}(1-t)\frac{n}{c}$ (notice $\sum_i m_i \leq \frac{n}{c}$) to get a partition $V_1, V_2, \ldots, V_w$ of the set of indices of buckets $[n/c]$ . Then set

$$S_i := \bigcup_{\ell \in V_i} B_\ell^{\neq 0},$$

where $B_\ell^{\neq 0}$ denotes the set of objects in bucket $B_\ell$ of class different from 0.

By construction we satisfy Item 1 of the lemma, namely that the $S_i$'s partition all the objects of class $\neq 0$. For the second item of the lemma, notice that for any interval $U \subseteq [n]$ (recall that $S_i^\chi$ is the set of objects in $S_i$ of class $\chi$)

$$|S_i^\chi \cap U| \geq \# \text{ buckets that compose } S_i$$

$$\text{and that are fully contained in } U,$$

since each bucket has 1 objects of class $\chi$. Let $J \subseteq [n/c]$ be the indices of the buckets fully contained in $U$; so the previous bound is

$$|S_i^\chi \cap U| \geq |V_i \cap J| \geq \frac{1}{2}\frac{|J|}{n/c}t^{i-1}(1-t)\frac{n}{c} - 12\log(n/c),$$

where the last inequality follows from Lemma 7. Moreover, we can see that $|J| \geq \frac{|U|}{c} - 2$. Replacing this bound in the previous displayed inequality and again using the fact $n \geq cst \cdot c$ for a sufficiently large $cst$ gives the desired result. $\qquad\square$

# D    Proof of Theorem 3

## D.1    The Bounds on the average case

In this section we show the bounds

$$\text{depth}_{avg}(T) \quad \leq \quad O\left(\log n\right)\text{depth}_{avg}^*, \tag{21}$$

$$\text{expl}_{avg}(T) \quad \leq \quad O\left(\log n\right)\text{expl}_{avg}^*. \tag{22}$$

for a tree $T$ produced by the algorithm SER-DT.

In Line 2 of SER-DT, the threshold factor $\frac{1}{2}$ is used to select an attribute that reduces the weighted pairwise misclassification. The argument we present here consider a more general case where $\frac{1}{2}$ is substituted by any $\gamma \in [\frac{1}{2}, 1)$. To simplify the argument, assume w.l.o.g. that (after the preprocessing) the weights are integers and $w_{\min} = 1$. To prove both parts of 21, it suffices to show that for the constant $\alpha := \max\{4, 2(\ln\frac{1}{\gamma})^{-1}\}$. we have

$$\text{depth}_{avg}(T) \leq \alpha \cdot \ln\left(P(O)w(O)\right) \cdot \text{depth}_{avg}(M^*), \tag{23}$$

where $M^*$ is a multiway decision tree[4] achieving the optimal average depth, $\text{depth}_{avg}^*$, for the instance $I$. Since $\ln(P(O)w(O)) \leq \ln(n^2 W) = O(\ln(nW))$, and using the equivalence in Lemma 3, this implies

$$\text{expl}_{avg}(T) \leq \text{depth}_{avg}(T) \leq O\left(\ln\left(nW\right)\right) \cdot \text{depth}_{avg}^*$$

$$= O\left(\ln\left(nW\right)\right) \cdot \text{expl}_{avg}^*.$$

---

[4]Recall that this is a decision tree where each test corresponds to an attribute $a$, and making this test splits the objects based on all possible values of the this attribute, instead of just splitting them as $a(o) < t$ and $a(o) \geq t$ for some threshold $t$.

To prove (23), we are going to use the following lower bound on the optimum average depth of a multiway tree.

**Lemma 8.** *Let $(a^*, i^*)$ be the pair attribute/value maximizing $\max_{a,i} P(O(a,i))w(O(a,i))$. Let $M^*$ be an optimal multiway tree, i.e., $\mathrm{depth}_{avg}(M^*) = \mathrm{depth}^*_{avg}$. Then*

$$\mathrm{depth}_{avg}(M^*) \geq \frac{1}{2} \frac{P(O)w(O)^2}{P(O)w(O) - P(O(a^*, i^*))w(O(a^*, i^*))}.$$

*Proof.* For a node $\nu$ of $M^*$, let $O_\nu$ be the set of objects associated with the leaves of the subtree rooted at $\nu$, and let $a_\nu$ be the attribute tested at node $\nu$. Like in (11) we compute the average cost of the optimal tree $M^*$ by summing up the weights of its internal nodes, $\mathrm{depth}_{avg}(M^*) = \sum_{\mathrm{int}(M^*)} w(O_\nu)$, where again we use $\mathrm{int}(\cdot)$ to denote the set of internal nodes of a tree.

For a set of objects $S \subseteq O$ and attribute $x$, define the quantity

$$\Delta_x(S) := \min_i \Big( P(S)w(S) - P(S(x,i))w(S(x,i)) \Big)$$
$$= P(S)w(S) - \max_i P(S(x,i))w(S(x,i)).$$

Notice that by definition, the attribute $x = a^*$ is the one that has the largest value $\Delta_x(O)$, that is, $\Delta_a(O) \leq \Delta_{a^*}(O)$ for every attribute $a$. Hence, we have

$$\mathrm{depth}_{avg}(M^*) = \sum_{\mathrm{int}(T^*)} w(O_\nu) \geq \frac{\sum_{\mathrm{int}(T^*)} w(O_\nu) \cdot \Delta_{a_\nu}(O)}{\Delta_{a^*}(O)}. \tag{24}$$

To make the numerator more "local", we would like to replace $\Delta_{a_\nu}(O)$ by $\Delta_{a_\nu}(O_\nu)$. The next lemma shows we can do this and still have a valid lower bound for $\mathrm{depth}_{avg}(M^*)$.

**Claim 1.** *For every attribute $x$, the function $\Delta_x(\cdot)$ is monotone, namely for any sets of objects $V \subseteq U \subseteq O$ we have $\Delta_x(V) \leq \Delta_x(U)$.*

*Proof of the claim.* Let $\bar{\imath} = \mathrm{argmin}_i (P(U)w(U) - P(U(x,i))w(U(x,i)))$ be the attribute value that yields the definition of $\Delta_x(U)$. Since $\bar{\imath}$ is a feasible value in the definition of $\Delta_x(V)$, we have $\Delta_x(V) \leq P(V)w(V) - P(V(x,\bar{\imath}))w(V(x,\bar{\imath}))$; so to show $\Delta_x(V) \leq \Delta_x(U)$ it suffices to show

$$P(V)w(V) - P(V(x,\bar{\imath}))w(V(x,\bar{\imath}))$$
$$\leq P(U)w(U) - P(U(x,\bar{\imath}))w(U(x,\bar{\imath})).$$

Let $S(x, \neq i) := S \setminus S(x,i)$ be the objects in $S$ with value different from $i$ in the attribute $x$, and notice $w(S) = w(S(x,i)) + w(S(x, \neq i))$. Applying this in the last displayed inequality, we see that it is equivalent to

$$P(V(x,\bar{\imath}))w(V(x, \neq \bar{\imath})) + \underbrace{\Big[ P(V) - P(V(x,\bar{\imath})) \Big]}_{\text{top-term}} \cdot w(V) \tag{25}$$

$$\leq P(U(x,\bar{\imath}))w(U(x, \neq \bar{\imath})) + \underbrace{\Big[ P(U) - P(U(x,\bar{\imath})) \Big]}_{\text{bottom-term}} \cdot w(U). \tag{26}$$

Since the number of pairs $P(S)$ and the weight $w(S)$ are monotone in $S$ and since $V \subseteq U$, it is clear that each term in (25) (except top-term) is at most the corresponding term in (26) (except bottom-term). Thus, to prove this inequality, it suffices to argue that top-term $\leq$ bottom-term. But notice that top-term is the number of pairs of objects $o \neq o' \in V$ that have different classes and where at least one of objects has value $\neq \bar{\imath}$ in attribute $x$; bottom-term is the same thing, but over pairs of objects $o \neq o'$ in $U$. Since every pair in the former also belongs to the latter, we see that top-term $\leq$ bottom-term. This then proves inequality (25)-(26) and concludes the proof of the claim. $\quad\square$

Applying this on (24) we obtain the lower bound

$$\text{depth}_{avg}(M^*) \geq \frac{\sum_{\text{int}(T^*)} w(O_\nu) \cdot \Delta_{a_\nu}(O_\nu)}{\Delta_{a^*}(O)}. \tag{27}$$

We can think of the numerator above as the cost of a decision tree for an instance $\tilde{I}$ with the same tests and object set as $I$ but with the additional property that applying the test on an attribute $a$ at a point where there are $S$ objects incurs a cost equal to $\Delta_a(S)$ In general, given a decision tree $\tilde{M}$ for the instance $\tilde{I}$, its cost is defined as

$$\sum_{\text{int}(\tilde{M})} w(\tilde{O}_\nu) \cdot \Delta_{\tilde{a}_\nu}(\tilde{O}_\nu),$$

where $\tilde{O}_\nu$ is the set of objects associated with the leaves of the subtree of $\tilde{M}$ rooted at $\nu$, and $\tilde{a}_\nu$ is the attribute tested at the node $\nu$. Letting $\text{depth}^*_{avg}(\tilde{I})$ denote the minimum cost of a multiway decision tree for $\tilde{I}$ with costs given by $\Delta$, from (27) we have

$$\text{depth}_{avg}(M^*) \geq \frac{\text{depth}^*_{avg}(\tilde{I})}{\Delta_{a^*}(O)}. \tag{28}$$

The proof of Lemma 8 is then completed by proving the following claim.

**Claim 2.** *We have* $\text{depth}^*_{avg}(\tilde{I}) \geq \frac{P(O)w(O)^2}{2}$.

*Proof of the claim.* We argue by induction on the number of pairs of distinct classes $P(O)$. For the base case $P(O) = 1$, we have that in an optimal tree the attribute $a$ tested at the root node $\rho$ must separate the two object belonging to the only pair — note that such an attribute must exists since we assume separability of objects belonging to different classes. Hence, it must hold that $P(O(a, i^a)) = 0$, and $\Delta_a(O) = P(O)w(O)$. Therefore we have

$$\text{depth}^*_{avg}(\tilde{I}) \geq w(O_\rho)\Delta_a(O) = P(O)w(O)^2,$$

as desired.

Now assume $P(O) \geq 2$ and that the claim holds for all instances with the number of pairs of distinct classes smaller than $P(O)$. Let $a$ be the attribute tested at the root of a tree attaining $\text{depth}^*_{avg}(\tilde{I})$. Also let $i^a$ be the value at attribute $a$ that yields $\Delta_a(O)$, namely $i^a = \text{argmin}_i(P(O)w(O) - P(O(a, i))w(O(a, i)))$. Because of the optimality of the tree we can assume that $P(O(a, i^a)) < P(O)$, i.e., the test $a$ separates at least one of the pairs.

Case 1. $P(O(a, i^a)) = 0$. Then as in the base case $\Delta_a(O) = P(O)w(O)$, and again we have $\text{depth}^*_{avg}(\tilde{I}) = w(O) \cdot \Delta_a(O) = P(O)w(O)^2$.

Case 2. $1 \leq P(S(a, i^a)) < P(S)$. Then we consider the contribution of the root node and the subtree rooted at the child corresponding to the value $i^a$. Let $\tilde{I}_{i^a}$ be the instance corresponding to the objects in the set $O(a, i^a)$, with number of pairs $P(O(a, i^a))$. We have

$$\text{depth}^*_{avg}(\tilde{I}) \geq w(O) \cdot \Delta_a(O) + \text{depth}^*_{avg}(\tilde{I}_{i^a}) \tag{29}$$

$$\geq w(O) \cdot \Big( P(O)w(O) - P(O(a, i^a))w(O(a, i^a)) \Big)$$

$$+ \frac{P(O(a, i^a))w(O(a, i^a))^2}{2} \tag{30}$$

$$\geq P(O)w(O)^2 - \frac{P(O(a, i^a))w(O)^2}{2} \tag{31}$$

$$\geq \frac{P(S)w(S)^2}{2}, \tag{32}$$

where (30)) follows from using the inductive hypothesis on $\text{depth}^*_{avg}(\tilde{I}_{i^a})$, (31) follows from the monotonicity of the weights $w(O(a, i^a)) \leq w(O)$, and (32) follows from monotonicity of the pairs $P(O(a, i^a)) \leq P(O)$. This concludes the proof of the inductive step, and hence of the claim. $\square$

$\square$

Given this lower bound on the average depth of an optimal multiway tree $\text{depth}_{avg}^*$ we now show the desired bound (23). Skipping trivialities, we can assume $n \geq 2$. We argue by induction on the number of pairs of distinct classes $P(O)$ in the instance . The base case $|P(O)| = 1$ is easily settled by the assumption that for every pair of objects from distinct classes there exists a test splitting them.

So assume by induction that (23) holds for the tree build by the algorithm on every instance $I'$ with objects $O'$ such that $P(O') < P(O)$. To prove that it still holds for the instance $I$, we consider whether the algorithm entered the If in Line 2 or not.

**Case 1: The algorithm enters the If at Line 2.** Let $(a, t)$ be the split used by the algorithm at the root. For $i = 1, 2$, let $T_i$ be the subtree of $T$ rooted at the left and right child of the root of $T$. Recall that $T_1$ is the tree built on the instance $I_1$ with object set $O_1 := O(a, \leq t)$ and $T_2$ is the the tree recursively built on the instance $I_2$ with object set $O_2 := O(a, > t)$. By construction we have $\texttt{wpm}(O(a, t)) \leq \gamma \cdot P(O)w(O)$, which means that

$$P(O_i)w(O_i) \leq \gamma P(O)w(O), \quad i = 1, 2. \tag{33}$$

The average depth of the tree $T$ build by the algorithm can then be decomposed as $\text{depth}_{avg}(T) = w(O) + \text{depth}_{avg}(T_1) + \text{depth}_{avg}(T_2)$. Moreover, we also have the following subadditivity property: $\text{depth}_{avg}^*(I) \geq \text{depth}_{avg}^*(I_1) + \text{depth}_{avg}^*(I_2)$. In fact, letting $M^*$ (resp. $M_1^*, M_2^*$) being a multiway tree achieving $\text{depth}_{avg}^*(I)$ (resp. $\text{depth}_{avg}^*(I_1)$, $\text{depth}_{avg}^*(I_2)$) we have

$$\text{depth}_{avg}^*(I) = \text{depth}_{avg}(M^*) = \sum_{o \in O} w(o) \cdot \text{depth}_{avg}^{T^*}(\ell(o))$$

$$\geq \sum_{o \in O_1} w(o) \cdot \text{depth}_{avg}^{M^*}(\ell(o)) + \sum_{o \in O_2} w(o) \cdot \text{depth}_{avg}^{M^*}(\ell(o))$$

$$\geq \sum_{o \in O_1} w(o) \cdot \text{depth}_{avg}^{M_1^*}(\ell(o)) + \sum_{o \in O_2} w(o) \cdot \text{depth}_{avg}^{M_2^*}(\ell(o))$$

$$= \text{depth}_{avg}^*(I_1) + \text{depth}_{avg}^*(I_2),$$

where we used the notation $\text{depth}_{avg}^M(\ell(o))$ to indicate the explanation size of leaf reached by object $o$ in a tree $M$. Then we have

$$\frac{\text{depth}_{avg}(T)}{\text{depth}_{avg}^*(I)} = \frac{w(O) + \text{depth}_{avg}(T_1) + \text{depth}_{avg}(T_2)}{\text{depth}_{avg}^*(I)}$$

$$\leq \frac{w(O)}{\text{depth}_{avg}^*(I)} + \frac{\text{depth}_{avg}(T_1) + \text{depth}_{avg}(T_2)}{\text{depth}_{avg}^*(I_1) + \text{depth}_{avg}^*(I_2)}$$

$$\leq \frac{w(O)}{\text{depth}_{avg}^*(I)} + \max_{i=1,2} \frac{\text{depth}_{avg}(T_i)}{OPT_M(I_i)}. \tag{34}$$

But $\text{depth}_{avg}^*(I) \geq w(O)$, and by the inductive hypothesis $\frac{\text{depth}_{avg}(T_i)}{\text{depth}_{avg}^*(I_i)} \leq \alpha \ln(P(O_i)w(O_i))$, which is at most $\alpha \ln(\gamma P(O)w(O))$ by (33). Plugging these bounds in the RHS of (34) gives

$$\frac{\text{depth}_{avg}(T)}{\text{depth}_{avg}^*(I)} \leq 1 + \alpha \ln \left( \gamma P(O)w(O) \right)$$

$$= \alpha \ln \left( P(O)w(O) \right) + \left( 1 - \alpha \ln \frac{1}{\gamma} \right) \leq \alpha \ln(P(O)w(O)),$$

where the last inequality follows from the fact $\alpha \geq (\ln \frac{1}{\gamma})^{-1}$. This proves the inductive hypothesis in this case.

**Case 2: The algorithm does not enter the If at Line 2.**  In this case, the top of the tree $T$ produced by the algorithm uses the splits $(a^*, t^* - 1)$ and $(a^*, t^*)$, and has (up to) three subtrees $T_1, T_2, T_3$ as children of these nodes, where:

- $T_1$ is constructed recursively for the instance $I_1$ with object set $O_1 := O(a^*, \leq t^* - 1)$
- $T_2$ is constructed recursively for the instance $I_2$ with object set $O_2 := O(a^*, t^*)$
- $T_3$ is constructed recursively for the instance $I_3$ with object set $O_3 := O(a^*, > t^*)$.

(Note that one of the instances $I_1, I_3$ may be empty.) The average depth of $T$ can be decomposed based on the contribution of the two splits $(a^*, t^* - 1)$ and $(a^*, t^*)$ (which is at most $2w(O)$) and of the subtrees $T_i$'s, namely

$$\mathrm{depth}_{avg}(T) \leq 2w(O) + \mathrm{depth}_{avg}(T_1) + \mathrm{depth}_{avg}(T_2)$$
$$+ \mathrm{depth}_{avg}(T_3).$$

Thus, as in inequality (34) we have

$$\frac{\mathrm{depth}_{avg}(T)}{\mathrm{depth}^*_{avg}(I)} \leq \frac{2w(O)}{\mathrm{depth}^*_{avg}(I)} + \max_{i=1,2,3} \frac{\mathrm{depth}_{avg}(T_i)}{\mathrm{depth}^*_{avg}(I_i)},$$

where if an instance $I_i$ is empty (so the tree $T_i$ does not exist) we ignore the corresponding term in the max.

By definition of $t^*$, for $i = 1, 3$ the instance $I_i$ satisfies $P(O_i)w(O_i) \leq \gamma P(O)w(O)$ , and so applying the induction hypothesis on the instances $I_i$ we get

$$\frac{\mathrm{depth}_{avg}(T)}{\mathrm{depth}^*_{avg}(I)} \leq \frac{2w(O)}{\mathrm{depth}^*_{avg}(I)}$$
$$+ \max \left\{ \alpha \ln \left( \gamma P(O)w(O) \right), \ \alpha \ln \left( P(O_2)w(O_2) \right) \right\}.$$

If $P(O_2)w(O_2) \leq \gamma P(O)w(O)$, the max becomes simply $\alpha \ln(\gamma P(O)w(O))$ and again using $\mathrm{depth}^*_{avg}(I) \geq w(O)$ we get

$$\frac{\mathrm{depth}_{avg}(T)}{\mathrm{depth}^*_{avg}(I)} \leq 2 + \alpha \ln \left( \gamma P(O)w(O) \right) \leq \alpha \ln(P(O)w(O)),$$

where the last inequality uses the fact $\alpha \geq 2(\ln \frac{1}{\gamma})^{-1}$; so the inductive hypothesis holds in this case.

Now consider the case where $P(O_2)w(O_2) > \gamma P(O)w(O)$, and hence the max becomes $\alpha \ln(P(O_2)w(O_2))$. In this case, we use the lower bound on $\mathrm{depth}^*_{avg}(I)$ from Lemma 8 to obtain (let $a^*$ and $i^*$ be defined as in the lemma)

$$\frac{\mathrm{depth}_{avg}(T)}{\mathrm{depth}^*_{avg}(I)} \leq \frac{4 \left( P(O)w(O) - P(O(a^*, i^*))w(O(a^*, i^*)) \right)}{P(O)w(O)}$$
$$+ \alpha \ln P(O_2)w(O_2)$$
$$\leq 4 \left( 1 - \frac{P(O(a^*, i^*))w(O(a^*, i^*))}{P(O)w(O)} \right)$$
$$+ \alpha \ln(P(O(a^*, i^*))w(O(a^*, i^*)))$$
$$\leq 4 \ln \frac{P(O)w(O)}{P(O(a^*, i^*))w(O(a^*, i^*))}$$
$$+ \alpha \ln(P(O(a^*, i^*))w(O(a^*, i^*)))$$
$$\leq \alpha \ln \left( P(S)w(S) \right),$$

where the second inequality follows the fact $P(O(a^*, i^*))w(O(a^*, i^*)) \geq P(O(a^*, t^*))w(a^*, t^*)) = P(O_2)w(O_2)$ (by optimality of $i^*$), the third inequality follows from the fact $1 - x \leq -\ln x$ valid for all $x > 0$ (since by convexity $\ln x \geq \ln(1) + \ln'(1)(x - 1)$), and the last inequality follows from the fact $\alpha \geq 4$.

This concludes the proof of the inductive step, and hence that inequality (23) holds and as a consequence, we have

$$\text{depth}_{avg}(T) \leq O\left(\ln\left(n\frac{W}{w_{\min}}\right)\right)\text{depth}^*_{avg},$$

$$\text{expl}_{avg}(T) \leq O\left(\ln\left(n\frac{W}{w_{\min}}\right)\right)\text{expl}^*_{avg},$$

where $W = w(O)$ is the total weight of the objects, $w_{\min} = \min_{o \in S} w(o)$ is the smallest weight of an object. Finally, by the preprocessing step (following the approach from [35]) we have that the approximation term $O\left(\ln\left(n\frac{W}{w_{\min}}\right)\right)$ becomes $O(\log n)$, i.e., we can remove the dependency on the weights and achieve the desired $O(\log n)$ approximation bound as given in the statement of the theorem.

### D.2 Proof of Worst Case

We now focus on the proof of the worst case bounds

$$\text{depth}_{wc}(T) \quad \leq \quad O\left(\log n\right)\text{depth}^*_{wc}, \tag{35}$$
$$\text{expl}_{wc}(T) \quad \leq \quad O\left(\log n\right)\text{expl}^*_{wc}. \tag{36}$$

for the decision tree $T$ built by SER-DT. Again to simplify the argument, assume w.l.o.g. that (after the preprocessing) the weights are integers and $w_{\min} = 1$.

Let $\mathcal{P}$ be longest path from the root to a leaf in $T$. For a node $g$ in $\mathcal{P}$, its weighted pair-wise misclassification (wpm) is the wpm of the set of objects that reach $g$. The nodes of $\mathcal{P}$ can be split based on whether they were added in Line 3 or Line 8 of one of the recursive calls of SER-DT. More specifically, let $G$ be the set of nodes in $\mathcal{P}$ that are associated with the set of objects $S(a^*, t^*)$ at Line 8 of SER-DT. We note that the total number of nodes in $\mathcal{P}$ is at most

$$\text{depth}_{wc}(T) = |\mathcal{P}| \leq 2|G| + \log(P(O)w(O)), \tag{37}$$

since in every recursive call where a node in $G$ is added to $\mathcal{P}$, its parent is also added, hence the factor of 2 in the formula; on the other hand, if the recursive call adds a node to $\mathcal{P}$ outside of $G$ (Line 3) then it means the condition in Line 2 was satisfied, and so the wpm of the node that is added to $\mathcal{P}$ is at most half of the wpm of its parent and, thus, only $O(\log(P(O)w(O))$ additions of this type are possible (recall that we are assuming (scaled) integer weights $\geq 1$).

Let $M^*_{wc}$ be a multiway decision tree achieving the minimum worst-case depth for the instance. We want to relate $|G|$ to $\text{depth}_{wc}(M^*_{wc})$. The following claim lower bounds the latter quantity and is the worst-case version of Lemma 8.

**Lemma 9.** *Consider a set $S \subseteq O$ of objects of the instance, and let $(a^*, i^*)$ be the pair attribute/value solving the optimization $\min_a \max_i P(S(a,i))w(S(a,i))$. Then*

$$\text{depth}_{wc}(M^*_{wc}) \geq \frac{1}{2}\frac{P(S)w(S)}{P(S)w(S) - P(S(a^*, i^*))w(S(a^*, i^*))}.$$

*Proof.* Define the normalized weights $w'(o) := \frac{w(o)}{w(S)}$, which then form a probability distribution over the objects in $S$. Let $M^*_{avg}$ be a multiway tree for $S$ with minimum average depth with respect to the weights $w'$, i.e. that—over all multiway decision tree $M$ for the instance—minimizes

$$\text{depth}_{avg}(M, w') = \sum_{\ell \in \text{leaf}(M)} w'(\ell) \cdot \text{depth}(\ell),$$

where we included $w'$ in $\text{depth}_{avg}(M, w')$ to avoid confusion.

Using the fact that the maximum is always at least the average (w.r.t. a probability distribution) and the optimality of $M^*_{avg}$ we get

$$\text{depth}_{wc}(M^*_{wc}) \geq \text{depth}_{avg}(M^*_{wc}, w') \geq \text{depth}_{avg}(M^*_{avg}, w'). \tag{38}$$

Moreover, applying Lemma 8 to the instance with objects $S$ and weights $w'$ we obtain

$$
\begin{aligned}
\mathrm{depth}_{avg}(M^*_{avg}, w') &\geq \frac{1}{2} \frac{P(S)w'(S)^2}{P(S)w'(S) - P(S(a^*, i^*))w'(S(a^*, i^*))} \\
&= \frac{1}{2} \frac{P(S)w(S)}{P(S)w(S) - P(S(a^*, i^*))w(S(a^*, i^*))},
\end{aligned} \tag{39}
$$

where the equality follows from using the fact $w'(S) = 1$ to remove one of these terms in the numerator and then multiplying both numerator and denominator by $w(S)$. (Observe that $(a^*, t^*)$ is exactly the one required in this application of Lemma 8 because the solutions to the problems $\min_a \max_i P(S(a, i))w(S(a, i))$ and $\min_a \max_i P(S(a, i))w'(S(a, i))$ are the same, as $w'$ is just a scaling of $w$.) Combining (38) and (39) gives the desired result. $\qquad \square$

We are now ready to upper bound $|G|$ with respect to the quantity in (39). For that, let $G = (g_1, g_2, \ldots, g_{|G|})$ be the nodes in $G$ in order of traversal of $\mathcal{P}$ (from root to leaf). For a node $g_j$, consider the call of SER-DT that created this node (in Line 8). Let $S_j$ be the set of objects that was received as input by this call, and let $(a^*_j, t^*_j)$ be the attribute and threshold value used in Line 8 to create node $g_j$. Thus, the objects of the whole instance that reach node $g_j$ (in the tree $T$) are precisely $S_j(a^*_j, t^*_j)$.

Applying Lemma 9 to the set of objects $S_j$ and reorganizing the terms gives

$$
\begin{aligned}
1 \leq\ & 2 \cdot \mathrm{depth}_{wc}(M^*_{wc}) \\
&\cdot \frac{P(S_j)w(S_j) - P(S_j(a^*_j, i^*_j))w(S_j(a^*_j, i^*_j))}{P(S_j)w(S_j)}.
\end{aligned}
$$

Thus:

$$
\begin{aligned}
|G| = {}& \sum_{j \leq |G|} 1 \\
\leq {}& 2 \cdot \mathrm{depth}_{wc}(M^*_{wc}) \\
& \cdot \sum_{j \leq |G|} \frac{P(S_j)w(S_j) - P(S_j(a^*, i^*))w(S_j(a^*, i^*))}{P(S_j)w(S_j)} \\
\leq {}& 2 \cdot \mathrm{depth}_{wc}(M^*_{wc}) \\
& \cdot \sum_{j \leq |G|} \ln\left( \frac{P(S_j)w(S_j)}{P(S_j(a^*_j, i^*_j))w(S_j(a^*_j, i^*_j))} \right),
\end{aligned}
$$

where the last step uses the inequality $1 - \frac{1}{x} \leq \ln x$ which is valid for all $x \in (0, 1]$. But by definition we have $S_{j+1} \subseteq S_j(a_j, t_j)$, and hence the denominator in the log can be lower bounded as $P(S_j(a^*_j, i^*_j))w(S_j(a^*_j, i^*_j)) \geq P(S_j)w(S_j)$, and hence we obtain the desired upper bound on $G$:

$$
\begin{aligned}
|G| \leq\ & 2 \cdot \mathrm{depth}_{wc}(M^*_{wc}) \cdot \sum_{j \leq |G|} \ln\left( \frac{P(S_j)w(S_j)}{P(S_{j+1})w(S_{j+1})} \right) \\
\leq\ & 2 \cdot \mathrm{depth}_{wc}(M^*_{wc}) \cdot \ln(P(S_1)w(S_1)) \\
\leq\ & 2 \cdot \mathrm{depth}_{wc}(M^*_{wc}) \cdot \ln(P(O)w(O)).
\end{aligned}
$$

Plugging this bound in (37) and using the equivalence between the optimal worst-case explanation size and the optimal worst-case depth for multiway trees from Lemma 2, i.e., $\mathrm{expl}^*_{wc} = \mathrm{depth}^*_{wc} = \mathrm{depth}_{wc}(M^*_{wc})$, we obtain

$$
\mathrm{depth}_{wc}(T) \leq 4\, \mathrm{expl}^*_{wc} \cdot \ln(P(O)w(O)) \leq O(\log n) \cdot \mathrm{expl}^*_{wc},
$$

the last inequality holding since $P(O) \leq n^2$ and, because of the pre-processing step before the algorithm we have $w(O) \leq n^4$. This implies both the bounds in (35)-(36), $\mathrm{expl}_{wc}(T) \leq O(\log n) \cdot \mathrm{expl}^*_{wc}$ and $\mathrm{depth}_{wc}(T) \leq O(\log n) \cdot \mathrm{depth}^*_{wc}$, thus completing the proof of the lemma.

# E   Experiments – Additional Details and Statistics

Our experiments were executed with the following settings: Notebook Inspiron 14-7460; Processor: 7th generation Intel Core i5; 16GB RAM (DDR4 at 2400 MHz); Storage type: SSD; GPU NVIDIA GeForce 940MX.

For our implementation we employed Python 3.8.10; pandas 1.4.2; numpy 1.22.3; sklearn 1.0.2. Our code is available on

```
https://github.com/user-anonymous-researcher/interpretable-dts
```

Our datasets as well as their main characteristics are described in Table 2.

Table 2: Datasets. $n$ is the number of examples; `numeric` the number of numerical features; `categ.` the number of categorical features; $d$ the total of features (after doing the one-hot encoding) and `classes` the number of output classes.

| Dataset | n | numeric | categ. | $d$ | classes | source |
|---|---|---|---|---|---|---|
| Anuran | 7195 | 22 | 0 | 22 | 4 | [33] |
| Audit Risk | 773 | 26 | 0 | 26 | 2 | [45] |
| Avila | 20867 | 10 | 0 | 10 | 12 | [53] |
| Banknote | 1372 | 4 | 0 | 4 | 2 | [6] |
| Bankruptcy Polish | 4885 | 64 | 0 | 64 | 2 | [54] |
| Cardiotocography | 2126 | 21 | 0 | 21 | 10 | [14] |
| Collins | 1000 | 19 | 0 | 19 | 30 | [55] |
| Defaults Credit Card | 30000 | 20 | 3 | 33 | 2 | [57] |
| Dry Bean | 13611 | 16 | 0 | 16 | 7 | [34] |
| EEG Eye State | 14980 | 14 | 0 | 14 | 2 | [51] |
| Htru2 | 17898 | 8 | 0 | 8 | 2 | [42] |
| Iris | 150 | 4 | 0 | 4 | 3 | [20] |
| Letter Recognition | 20000 | 16 | 0 | 16 | 26 | [38] |
| Mice | 552 | 77 | 3 | 83 | 8 | [27] |
| OBS Network | 1060 | 19 | 2 | 24 | 4 | [50] |
| Occupancy Room | 10129 | 16 | 0 | 16 | 4 | [49] |
| Online Shoppers Intention | 12330 | 12 | 5 | 54 | 2 | [52] |
| Pen Digits | 10992 | 16 | 0 | 16 | 10 | [5] |
| Poker Hand | 1025010 | 10 | 0 | 10 | 10 | [15] |
| Sensorless | 58509 | 48 | 0 | 48 | 11 | [7] |

To generate the plots from Figure 2, we executed SER-DT with `FactorExpl` varying in the set

$$\{0.1 \times i | i = 1, \dots, 9\} \cup \{0.9 + 0.01 \times i | i = 1, \dots, 10\}.$$

In the following sections, we present some additional metrics and visualizations related to the experiments described in Section 6. Moreover, we also present two new experiments: in one of them, we remove the constraint on the depth of the trees while in the other we replace the `Gini` with entropy as a splitting criterion. We note that each entry presented in the following tables is given by the the average of 10 runs using different seeds to select the examples in the training and testing set.

## E.1   Metrics related to depth and running time

Table 3 shows the metrics $\text{depth}_{avg}$ and $\text{depth}_{wc}$ for the decision trees produced by both `CART` and SER-DT (experiment from Section 6). The metric $\text{depth}_{avg}$ for our algorithm is better than that of `CART` for 17 datasets and it is worse for only 3. For $\text{depth}_{wc}$ the results are similar, which is somehow expected since we set the maximum allowed depth to 6.

Table 4 shows the average running times in seconds of `CART` and our algorithm. The ratio between the fastest and slowest is at most 2 for all datasets.

Table 3: $\text{depth}_{avg}$ and $\text{depth}_{wc}$ for the experiment described in Section 6. We bold-faced the cases where the difference in accuracy is larger than $1\%$ and also the cases where the gain in terms of $\text{depth}_{avg}$ or $\text{depth}_{wc}$ is larger than $25\%$.

| Dataset | Test Accuracy | | $\text{depth}_{avg}$ | | $\text{depth}_{wc}$ | |
|---|---|---|---|---|---|---|
| | SER-DT | CART | SER-DT | CART | SER-DT | CART |
| anuran | 94,8% | 94,7% | 5,38 | 5,57 | 6,00 | 6,00 |
| audit risk | 99,9% | 99,9% | 1,00 | 1,00 | 1,00 | 1,00 |
| avila | 61,5% | **63,2%** | 5,14 | 5,29 | 6,00 | 6,00 |
| banknote | 97,6% | 98,1% | 4,39 | 4,43 | 6,00 | 6,00 |
| bankruptcy polish | 96,6% | 96,9% | 4,45 | 4,94 | 6,00 | 6,00 |
| cardiotocography | 89,5% | 89,8% | 4,98 | 5,51 | 6,00 | 6,00 |
| collins | 13,2% | **15,6%** | 5,89 | 5,91 | 6,00 | 6,00 |
| default credit card | 82,0% | 81,9% | **2,15** | 4,33 | 6,00 | 6,00 |
| dry bean | 90,1% | 89,8% | 4,76 | 5,41 | 6,00 | 6,00 |
| eeg eye state | 74,1% | 73,6% | 5,15 | 5,47 | 6,00 | 6,00 |
| htru2 | 97,7% | 97,7% | **2,80** | 5,30 | 6,00 | 6,00 |
| iris | 94,2% | 93,6% | 2,50 | 2,52 | 4,90 | 4,80 |
| letter recognition | 44,9% | **47,9%** | 5,96 | 5,94 | 6,00 | 6,00 |
| mice | 99,9% | 99,9% | 3,05 | 3,05 | 3,60 | 3,60 |
| obs network | **91,7%** | 89,5% | 4,47 | 4,39 | 6,00 | 6,00 |
| occupancy room | 99,4% | 99,3% | 4,72 | 4,83 | 6,00 | 6,00 |
| online shoppers intention | 89,3% | 89,8% | 3,89 | 4,77 | 6,00 | 6,00 |
| pen digits | **88,6%** | 86,9% | 5,73 | 5,80 | 6,00 | 6,00 |
| poker hand | 52,9% | **55,0%** | 4,61 | 4,30 | 6,00 | 5,30 |
| sensorless | **87,4%** | 80,1% | 5,26 | 5,33 | 6,00 | 6,00 |
| Average | 82,3% | 82,2% | 4,31 | 4,70 | 5,58 | 5,54 |

Table 4: Running times for CART and our algorithm for the experiment described in Section 6

| Dataset | SER-DT (sec) | CART (sec) |
|---|---|---|
| anuran | 7.0 | 3.7 |
| audit risk | 0.1 | 0.1 |
| avila | 9.0 | 9.2 |
| banknote | 0.2 | 0.1 |
| bankruptcy polish | 9.8 | 5.2 |
| cardiotocography | 1.4 | 1.7 |
| collins | 2.2 | 1.7 |
| default credit card | 18.1 | 19.3 |
| dry bean | 12.1 | 7.1 |
| eeg eye state | 3.5 | 4.0 |
| htru2 | 5.0 | 3.0 |
| iris | 0.0 | 0.0 |
| letter recognition | 17.1 | 26.1 |
| mice | 1.7 | 1.0 |
| obs network | 0.5 | 0.5 |
| occupancy room | 2.7 | 3.7 |
| online shoppers intention | 9.1 | 11.7 |
| pen digits | 5.4 | 7.1 |
| poker hand | 330.8 | 443.6 |
| sensorless | 178.5 | 120.5 |

### E.2 Sensitivity to FactorExpl

Figure 2 suggests that FactorExpl can be used to provide a trade-of between accuracy and explainability (measured according to $\text{expl}_{avg}$). To provide additional evidence, in Table 5 we show the test accuracy and $\text{expl}_{avg}$ for FactorExpl $\in \{0.9, 0.95, 0.99\}$.

Table 5: Sensitivity with respect to `FactorExpl`(FE below)

| Dataset | Test Accuracy | | | $\text{expl}_{avg}$ | | |
|---|---|---|---|---|---|---|
| | FE= 0,99 | FE=0,95 | FE=0,90 | FE=0,99 | FE=0,95 | FE=0,90 |
| anuran | 94,8% | 94,8% | 94,6% | 5,11 | 4,50 | 3,85 |
| audit risk | 99,9% | 99,9% | 99,9% | 1,00 | 1,00 | 1,00 |
| avila | 66,0% | 57,8% | 55,9% | 4,22 | 1,99 | 1,64 |
| banknote | 97,8% | 97,6% | 97,5% | 2,51 | 2,38 | 2,36 |
| bankruptcy polish | 96,7% | 96,6% | 96,7% | 3,73 | 1,92 | 1,50 |
| cardiotocography | 89,7% | 89,3% | 89,5% | 4,51 | 4,10 | 3,98 |
| collins | 15,9% | 13,1% | 12,8% | 3,74 | 1,61 | 1,35 |
| default credit card | 81,9% | 82,0% | 82,0% | 1,94 | 1,41 | 1,40 |
| dry bean | 89,7% | 90,0% | 89,8% | 3,60 | 3,25 | 3,05 |
| eeg eye state | 74,6% | 72,9% | 67,1% | 4,32 | 3,15 | 1,82 |
| htru2 | 97,7% | 97,7% | 97,6% | 1,27 | 1,19 | 1,15 |
| iris | 94,2% | 94,2% | 94,2% | 1,75 | 1,75 | 1,75 |
| letter recognition | 51,9% | 42,4% | 40,1% | 4,20 | 2,90 | 2,59 |
| mice | 99,9% | 99,9% | 99,9% | 3,05 | 3,05 | 3,05 |
| obs network | 91,7% | 91,3% | 90,3% | 3,75 | 3,15 | 2,67 |
| occupancy room | 99,3% | 99,4% | 99,5% | 4,25 | 4,15 | 4,07 |
| online shoppers intention | 89,9% | 89,1% | 88,7% | 3,82 | 3,12 | 2,94 |
| pen digits | 88,5% | 88,6% | 87,7% | 4,90 | 4,75 | 4,42 |
| poker hand | 53,0% | 52,8% | 52,7% | 1,89 | 1,79 | 1,76 |
| sensorless | 87,5% | 87,0% | 85,9% | 3,01 | 2,85 | 2,67 |
| Average | 83,0% | 81,8% | 81,1% | 3,33 | 2,70 | 2,45 |

We observe that $\text{expl}_{avg}$ has a very predictable behavior: the smaller the `FactorExpl` the smaller the $\text{expl}_{avg}$ (last 3 columns of this table). In terms of accuracy, for 13 datasets the difference between `FactorExpl` $= 0.99$ and `FactorExpl` $= 0.9$ is smaller than $1\%$. This number increases to 16 when we consider `FactorExpl` $= 0.99$ and `FactorExpl` $= 0.95$. We do not recommend using small values for `FactorExpl`($< 0.9$) in practice because the loss in terms of accuracy may be severe. In fact, we recommend using SER-DT with `FactorExpl` $\in [0.95, 0.99]$.

### E.3 Removing the maximum depth constraint

In our experiments, in order to prevent (very) large trees, we set the maximum allowed depth to 6. Table 6 and 7 show the results when this constraint is removed. Some observations are in order:

- SER-DT builds trees that are much more shallower than those built by `CART` (Columns 6 and 7 of Table 7). This was not possible to observe when the maximum depth was set to 6;
- The gains of SER-DT over `CART` for all explainability metrics become larger when the depth constraint is removed.

### E.4 Entropy as a split criterion

In our experiments on Section 6 we compared SER-DT against `CART`. We also implemented a variation of our method that employs the Shannon Entropy, rather than `Gini`, to evaluate the goodness of a split. We note that (a normalized version of) the entropy is employed by the widely used `C4.5` algorithm [48]. We tested our variation against a variation of `CART` that uses the entropy rather than `Gini` to evaluate the quality of a split and also to determine whether a leaf shall be expanded or not (stopping rule).

Table 8 presents the results for the test accuracy and the metrics related to the explanation size. The results are inline with those from Section 6.

Table 6: Test accuracy, $\mathrm{expl}_{avg}$ and $\mathrm{expl}_{wc}$ when no limit on the maximum depth is set. We bold-faced the cases where the difference in accuracy is larger than $1\%$ and also the cases where the gain in terms of $\mathrm{expl}_{avg}$ or $\mathrm{expl}_{wc}$ is larger than $30\%$.

| Dataset | Test Accuracy | | $\mathrm{expl}_{avg}$ | | $\mathrm{expl}_{wc}$ | |
|---|---|---|---|---|---|---|
| | SER-DT | CART | SER-DT | CART | SER-DT | CART |
| anuran | 95,7% | 95,6% | **6,00** | 8,74 | 9,4 | 11,1 |
| audit risk | 99,9% | 99,9% | 1,00 | 1,00 | 1 | 1 |
| avila | 92,8% | **98,4%** | 5,47 | 5,81 | 9 | 9,6 |
| banknote | 97,8% | 98,0% | 2,46 | 2,56 | 3,8 | 3,8 |
| bankruptcy polish | 95,9% | 95,8% | **3,14** | 7,25 | **8** | 13 |
| cardiotocography | 87,6% | 88,0% | **5,73** | 8,54 | **8,7** | 14,6 |
| collins | 12,0% | **14,5%** | **4,30** | 7,81 | **7,3** | 12,7 |
| default credit card | 72,7% | 72,5% | **5,74** | 13,08 | **11,1** | 20,2 |
| dry bean | 89,5% | 89,3% | **4,60** | 7,41 | 8,6 | 11,4 |
| eeg eye state | 82,2% | 82,9% | 6,48 | 8,31 | 11,1 | 13 |
| htru2 | 96,7% | 96,7% | **1,74** | 4,24 | 6,2 | 7,3 |
| iris | 94,2% | 93,6% | 1,75 | 1,76 | 3,1 | 3,4 |
| letter recognition | 84,3% | **86,2%** | 6,84 | 8,83 | 11,5 | 14,8 |
| mice | 99,9% | 99,9% | 3,05 | 3,05 | 3,6 | 3,6 |
| obs network | 100,0% | 100,0% | 3,94 | 4,83 | 6,5 | 8,2 |
| occupancy room | 99,6% | 99,5% | 4,26 | 4,98 | 6,9 | 7,9 |
| online shoppers intention | 86,2% | 86,3% | **5,72** | 9,15 | **10,1** | 16,4 |
| pen digits | 95,7% | 96,0% | 6,64 | 7,87 | 10,4 | 12,7 |
| poker hand | **81,0%** | 62,5% | **6,05** | 8,74 | 10 | 10 |
| sensorless | 98,4% | 98,1% | **4,93** | 9,60 | **10,3** | 20,3 |
| Average | 88,1% | 87,7% | 4,49 | 6,68 | 7,83 | 10,75 |

Table 7: Test accuracy, $\mathrm{depth}_{avg}$ and $\mathrm{depth}_{wc}$ when no limit on the maximum depth is set. We bold-faced the cases where the difference in accuracy is larger than $1\%$ and also the cases where the gain in terms of $\mathrm{depth}_{avg}$ or $\mathrm{depth}_{wc}$ is larger than $20\%$.

| Dataset | Test Accuracy | | $\mathrm{depth}_{avg}$ | | $\mathrm{depth}_{wc}$ | |
|---|---|---|---|---|---|---|
| | SER-DT | CART | SER-DT | CART | SER-DT | CART |
| anuran | 95,7% | 95,6% | **7,69** | 10,79 | **11,2** | 15,1 |
| audit risk | 99,9% | 99,9% | 1,00 | 1,00 | 1,0 | 1,0 |
| avila | 92,8% | **98,4%** | 11,25 | 11,49 | 18,9 | 21,1 |
| banknote | 97,8% | 98,0% | 4,51 | 4,60 | 6,9 | 7,3 |
| bankruptcy polish | 95,9% | 95,8% | **5,37** | 8,22 | **9,8** | 16,6 |
| cardiotocography | 87,6% | 88,0% | **7,56** | 10,20 | **11,5** | 17,5 |
| collins | 12,0% | **14,5%** | 9,44 | 10,55 | **16,2** | 21,1 |
| default credit card | 72,7% | 72,5% | **12,46** | 19,89 | **21,0** | 42,3 |
| dry bean | 89,5% | 89,3% | **9,20** | 12,16 | **15,8** | 25,0 |
| eeg eye state | 82,2% | 82,9% | 10,81 | 13,38 | **18,1** | 25,2 |
| htru2 | 96,7% | 96,7% | **8,26** | 10,86 | **14,4** | 22,2 |
| iris | 94,2% | 93,6% | 2,50 | 2,52 | 4,9 | 4,8 |
| letter recognition | 84,3% | **86,2%** | 11,39 | 12,92 | 22,4 | 27,8 |
| mice | 99,9% | 99,9% | 3,05 | 3,05 | 3,6 | 3,6 |
| obs network | 100,0% | 100,0% | 5,36 | 5,73 | 8,9 | 9,8 |
| occupancy room | 99,6% | 99,5% | 5,09 | 6,08 | 9,6 | 10,5 |
| online shoppers intention | 86,2% | 86,3% | **9,01** | 11,56 | **15,8** | 24,2 |
| pen digits | 95,7% | 96,0% | 8,91 | 9,73 | 14,1 | 17,6 |
| poker hand | **81,0%** | 62,5% | 17,92 | 20,75 | **30,5** | 40,7 |
| sensorless | 98,4% | 98,1% | **9,64** | 13,66 | **17,6** | 31,6 |
| Average | 88,1% | 87,7% | 8,02 | 9,96 | 13,6 | 19,3 |

Table 8: $\text{expl}_{avg}$ and $\text{expl}_{wc}$ when Entropy (Entr.) is used instead of `GINI`. We bold-faced the cases where the difference in accuracy is larger than $1\%$ and also the cases where the gain in terms of $\text{expl}_{avg}$ or $\text{expl}_{wc}$ is larger than $20\%$.

| Dataset | Test Accuracy | | $\text{expl}_{avg}$ | | $\text{expl}_{wc}$ | |
|---|---|---|---|---|---|---|
| | SER-DT $_{Ent}$ | Entr. | SER-DT $_{Ent}$ | Entr. | SER-DT $_{Ent}$ | Entr. |
| anuran | 94,5% | 94,5% | 4,92 | 5,06 | 6,0 | 6,0 |
| audit risk | 99,9% | 99,9% | 1,00 | 1,00 | 1,0 | 1,0 |
| avila | **66,5%** | 65,3% | 3,54 | 4,18 | 5,0 | 5,1 |
| banknote | 97,9% | 97,9% | 2,41 | 2,51 | 3,8 | 3,8 |
| bankruptcy polish | 97,1% | 97,3% | 2,95 | 3,17 | 5,6 | 6,0 |
| cardiotocography | 89,1% | 89,6% | 4,36 | 4,69 | 6,0 | 6,0 |
| collins | 14,1% | 14,2% | **3,51** | 4,81 | 5,0 | 5,7 |
| default credit card | 82,0% | 81,9% | **1,41** | 3,82 | **4,2** | 6,0 |
| dry bean | 90,1% | 90,0% | 3,66 | 4,34 | 5,8 | 6,0 |
| eeg eye state | **73,2%** | 72,1% | 3,42 | 4,26 | 5,6 | 6,0 |
| htru2 | 97,8% | 97,8% | **1,47** | 1,97 | 4,6 | 5,0 |
| iris | 94,5% | 94,2% | 1,70 | 1,73 | 3,2 | 3,3 |
| letter recognition | 58,5% | **59,6%** | 4,13 | 5,01 | 6,0 | 6,0 |
| mice | 99,9% | 99,9% | 3,00 | 3,00 | 3,0 | 3,0 |
| obs network | **89,4%** | 87,5% | 3,54 | 4,17 | 5,2 | 6,0 |
| occupancy room | 99,5% | 99,5% | 3,27 | 3,37 | 4,6 | 5,0 |
| online shoppers intention | 89,1% | 89,9% | **2,63** | 3,47 | **4,2** | 5,9 |
| pen digits | 89,6% | 89,7% | 5,12 | 5,44 | 6,0 | 6,0 |
| poker hand | 53,0% | **55,7%** | **1,83** | 4,53 | **4,0** | 5,0 |
| sensorless | 90,0% | 90,0% | **3,08** | 4,53 | 4,9 | 5,8 |
| Average | 83,3% | 83,3% | 3,05 | 3,71 | 4,68 | 5,13 |

### E.5 Boxplots for CART and SER-DT

Figures 5-8 present boxplots for the test accuracy of CART and SER-DT for the experiments described on Section 6. Figures 9-12 present boxplots for the $\text{expl}_{avg}$.

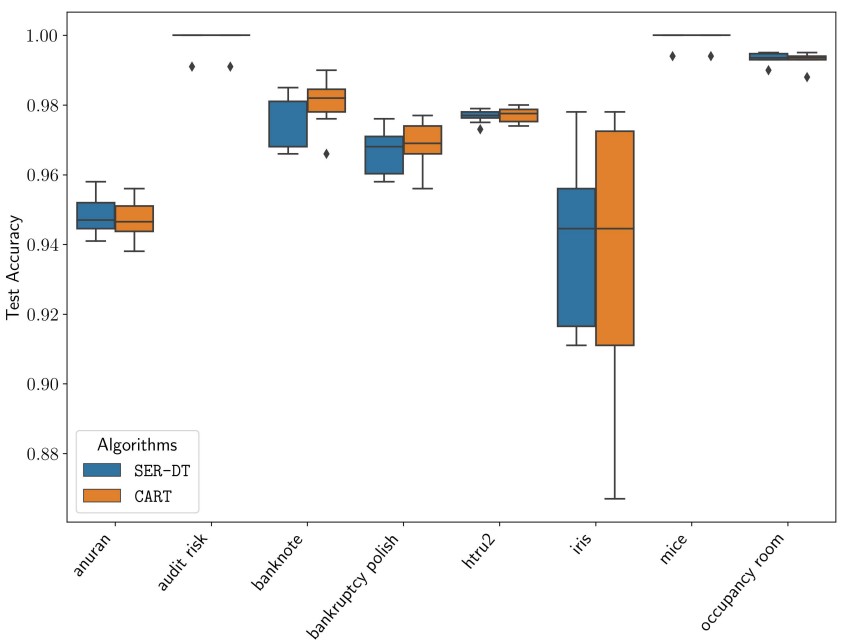

Figure 5: Test Accuracy for CART and SER-DT for some datasets

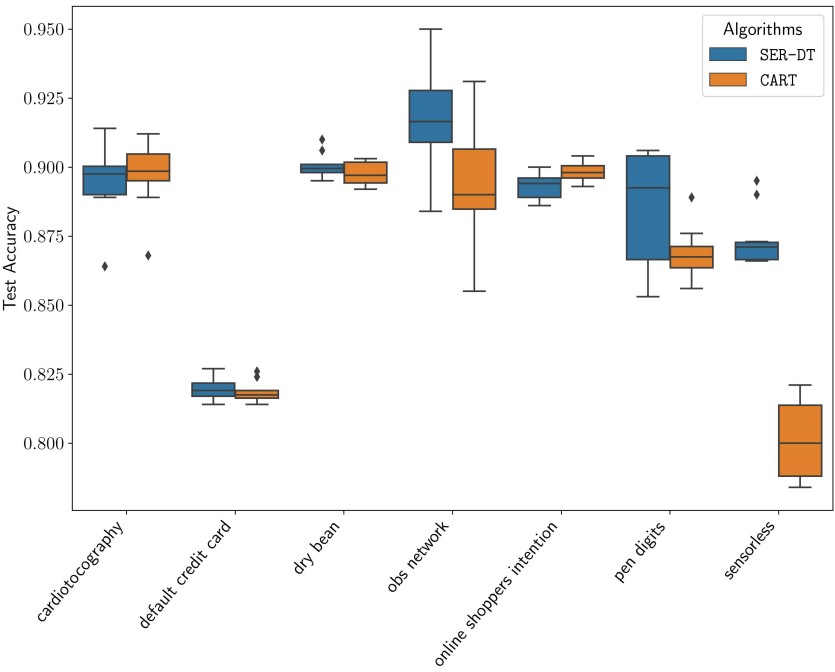

Figure 6: Test Accuracy for CART and SER-DT for some datasets

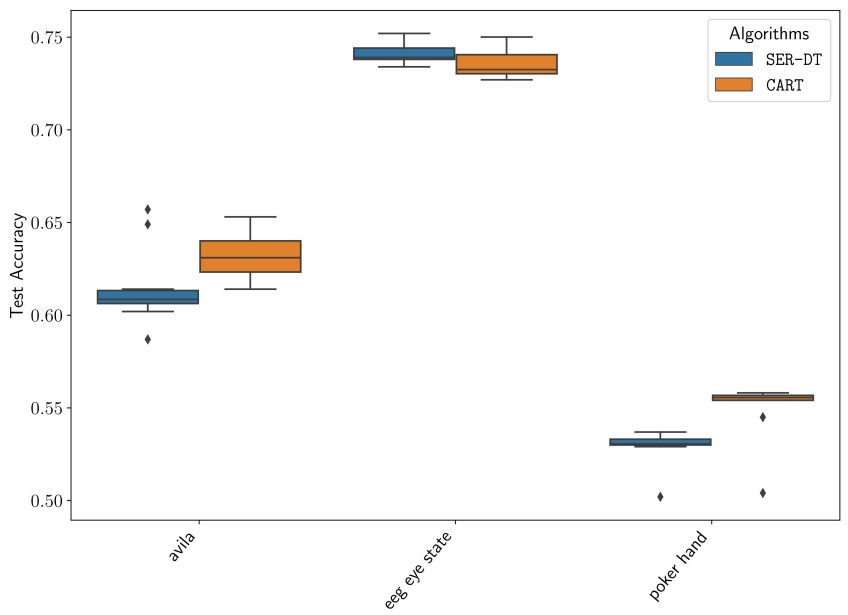

Figure 7: Test Accuracy for CART and SER-DT for some datasets

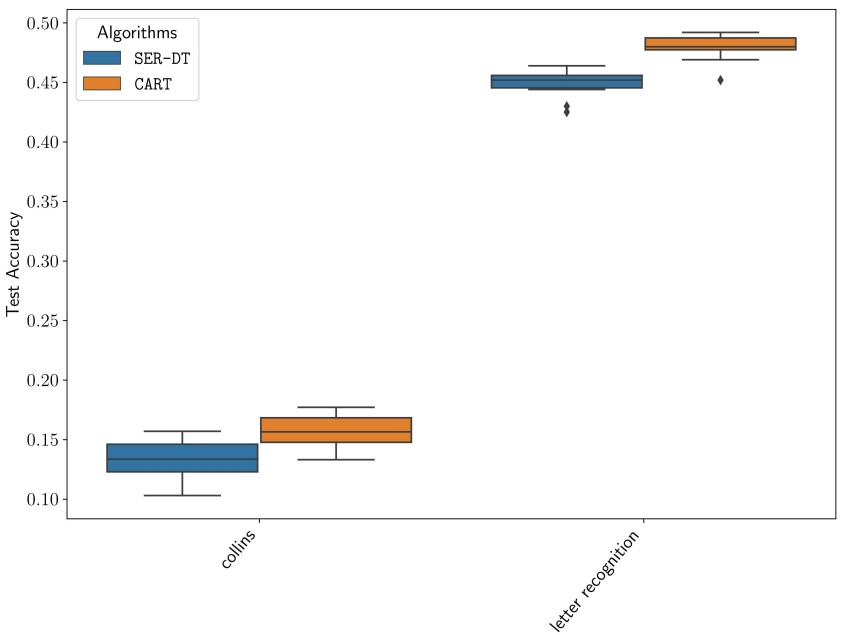

Figure 8: Test Accuracy for CART and SER-DT for some datasets

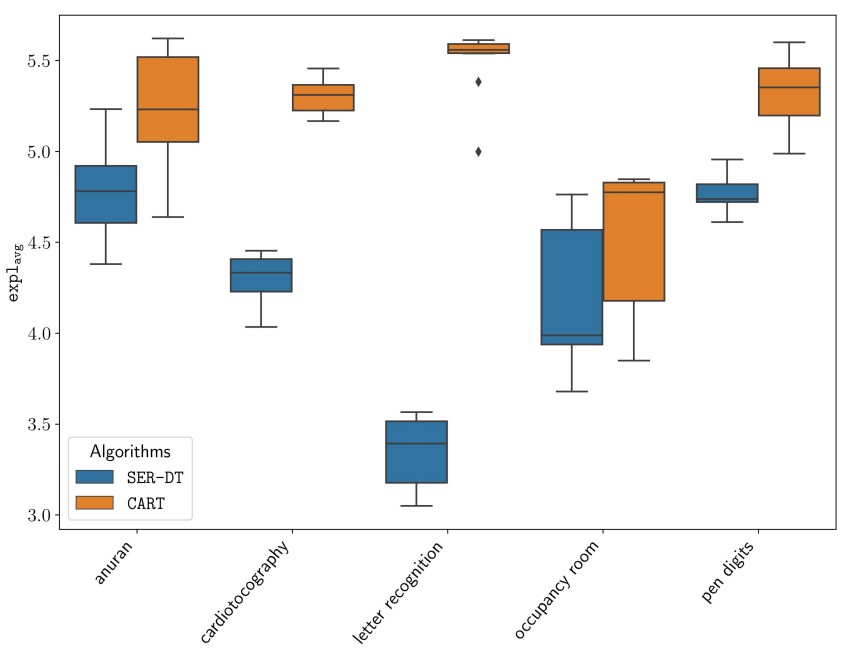

Figure 9: $\mathrm{expl}_{avg}$ for CART and SER-DT for some datasets

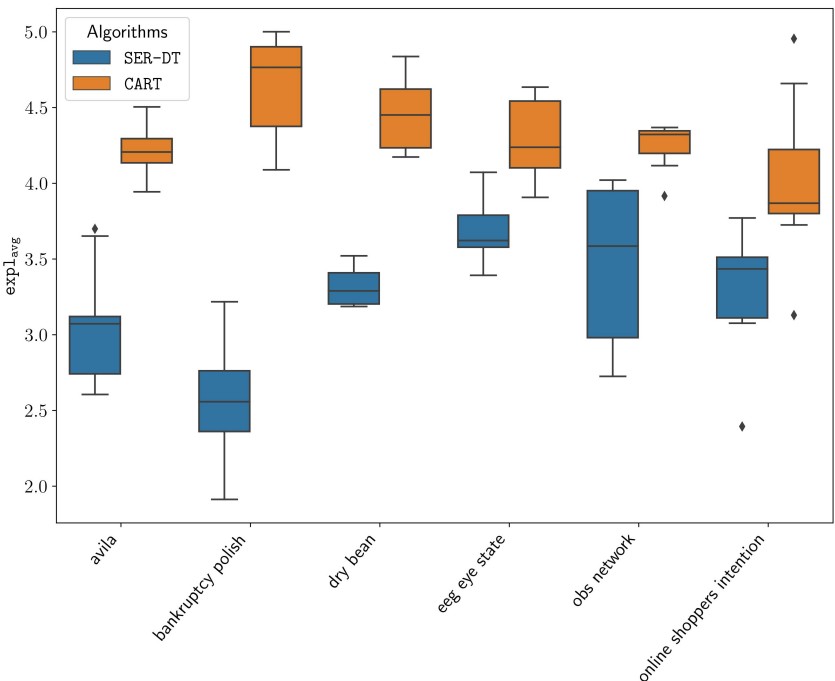

Figure 10: $\mathrm{expl}_{avg}$ for CART and SER-DT for some datasets

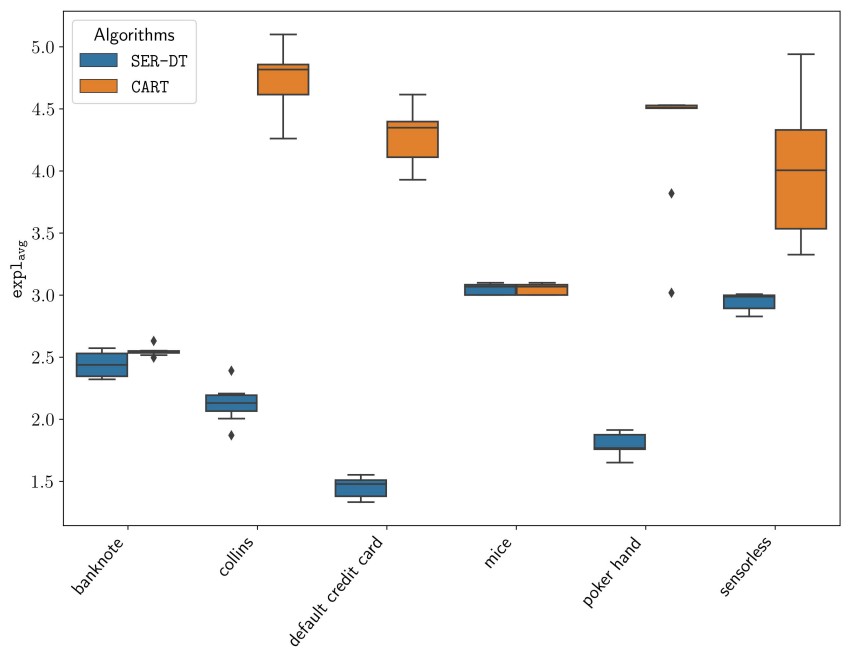

Figure 11: $\mathrm{expl}_{avg}$ for CART and SER-DT for some datasets

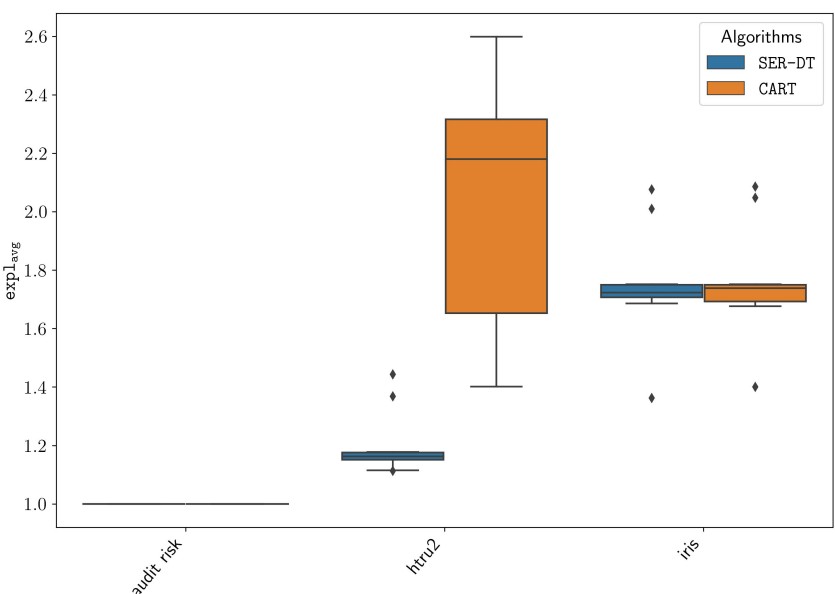

Figure 12: $\mathrm{expl}_{avg}$ for CART and SER-DT for some datasets

### E.6 Other examples of trees

We present some other examples of trees produced by SER-DT and `CART`. These examples provide additional evidence that our algorithm generates trees with better explainability but similar accuracy compared to `CART`. All trees were constructed by setting a maximum allowed depth to 4. For SER-DT we set `FactorExpl` $= 0.97$, such as reported on Section 6.

Our first example employs dataset `default credit card`. Figures 13 and 14 show the trees produced by `CART` and SER-DT, respectively. `CART` has a Test Accuracy of $81.9\%$, while SER-DT obtains $81.8\%$. In contrast, visually the SER-DT tree is much simpler than that of `CART`, and SER-DT achieves $\text{expl}_{avg} = 1.19$ against $\text{expl}_{avg} = 2.27$ from `CART`.

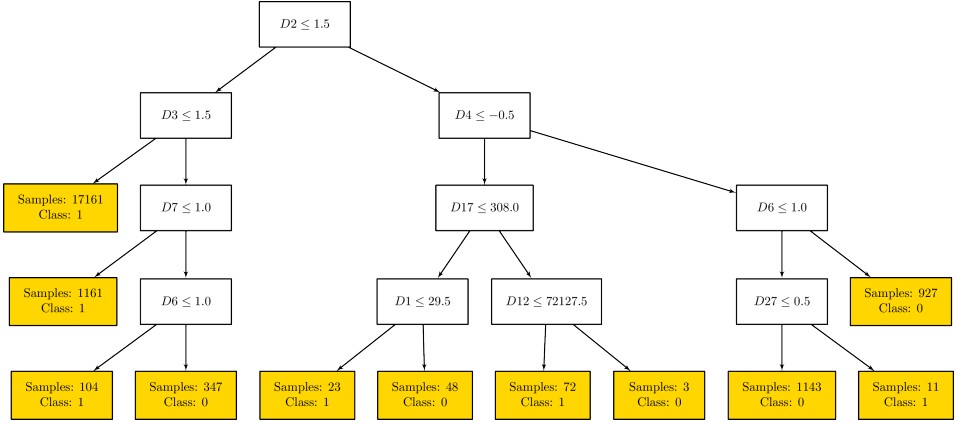

Figure 13: Tree produced with `CART` for dataset `default credit cart`

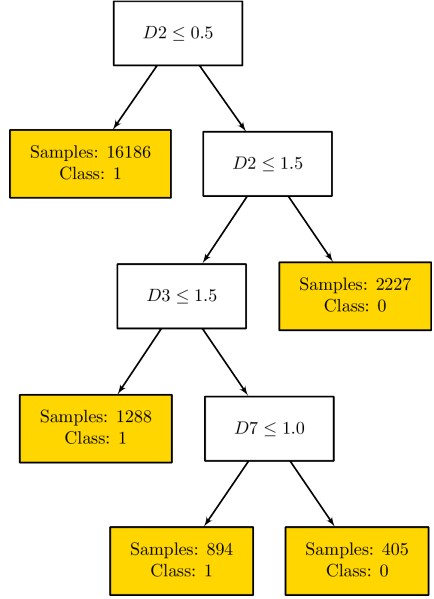

Figure 14: Tree produced with SER-DT for dataset `default credit cart`
.

Our second example uses dataset `online shoppers intention`, whose corresponding decision trees are represented in Figures 15 and 16. `CART` generates a tree with a Test Accuracy of $90.0\%$, while SER-DT achieves $89.1\%$ for this metric. In this case, SER-DT also clearly yields a simpler tree, achieving $\text{depth}_{avg} = 1.54$ against $\text{depth}_{avg} = 2.38$ from `CART`.

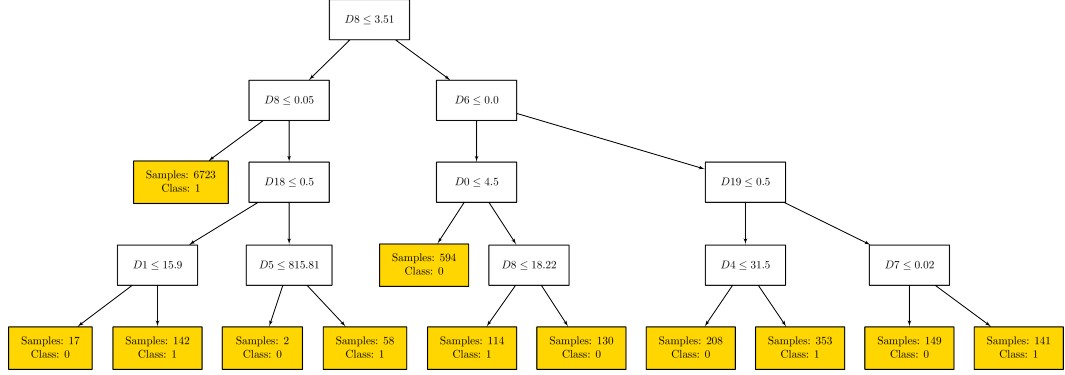

Figure 15: Tree produced with `CART` for dataset `online shoppers intention`

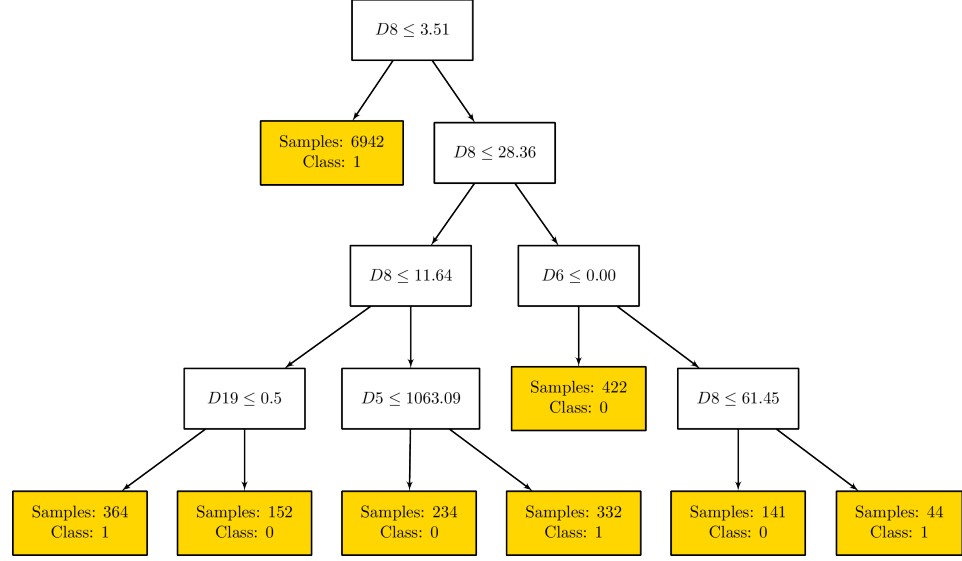

Figure 16: Tree produced with SER-DT for dataset `online shoppers intention`

The final example shows trees for dataset `dry bean`, in Figures 17 and 18. Although at first glance there doesn't seem to be much improvement, SER-DT yields a tree with $\text{expl}_{avg}$ of $2.60$, in contrast to $3.49$ for `CART`, which is a reduction of $25.6\%$. In terms of prediction, SER-DT gives $82.8\%$ of Test Accuracy, while `CART` gives $82.4\%$.

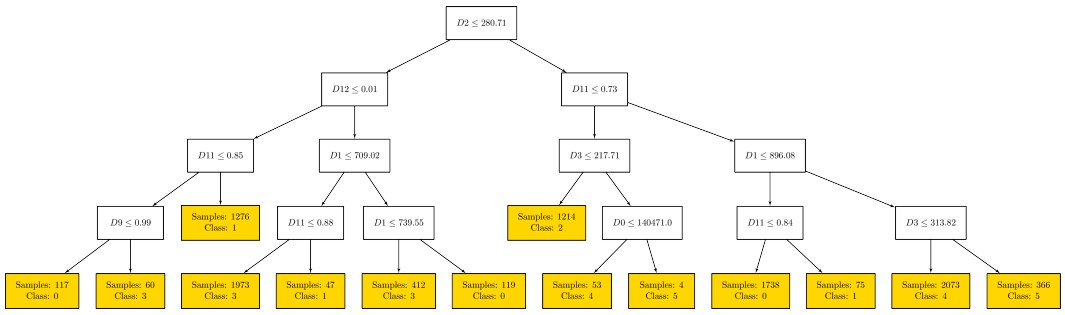

Figure 17: Tree produced with `CART` for dataset `dry bean`

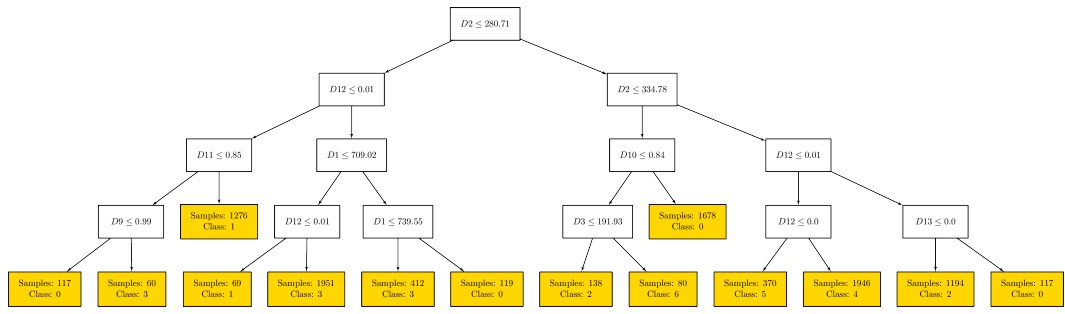

Figure 18: Tree produced with SER-DT for dataset `dry bean`

Table 9: Test Accuracy, $\text{expl}_{avg}$ and $\text{expl}_{wc}$ for `FactorExpl` $= 0.97$. Each entry is the average of 10 runs using different seeds to select the examples in the training and testing set.

| Dataset | Test Accuracy | | $\text{expl}_{avg}$ | | $\text{expl}_{wc}$ | |
|---|---|---|---|---|---|---|
| | SER-DT | EC$^2$ | SER-DT | EC$^2$ | SER-DT | EC$^2$ |
| anuran | **94.8 ± 0.5%** | 89.4 ± 1.7% | 4.78 | 4.36 | 6.00 | 5.90 |
| audit risk | 99.9 ± 0.3% | 99.3 ± 0.8% | **1.00** | 3.07 | **1.00** | 3.90 |
| avila | **61.5 ± 2.1%** | 57.7 ± 1.2% | 3.06 | 4.03 | 4.90 | 6.00 |
| banknote | **97.6 ± 0.8%** | 92.3 ± 1.7% | 2.44 | 1.88 | 3.80 | 3.00 |
| bankruptcy polish | 96.6 ± 0.7% | 97.4 ± 0.4% | 2.56 | 2.19 | 5.60 | 5.50 |
| cardiotocography | **89.5 ± 1.3%** | 80.1 ± 3.3% | 4.30 | 3.81 | 5.90 | 5.70 |
| collins | 13.2 ± 1.8% | **16.0 ± 1.2%** | **2.13** | 4.95 | **4.40** | 6.00 |
| default credit card | **82.0 ± 0.4%** | 80.5 ± 0.3% | **1.45** | 3.65 | **4.50** | 6.00 |
| dry bean | **90.1 ± 0.4%** | 83.9 ± 0.5% | 3.32 | 4.40 | 5.10 | 6.00 |
| eeg eye state | **74.1 ± 0.5%** | 71.1 ± 0.5% | 3.69 | 3.78 | 5.90 | 5.60 |
| htru2 | 97.7 ± 0.2% | 97.7 ± 0.1% | 1.20 | 1.52 | 4.30 | 4.60 |
| iris | **94.2 ± 2.6%** | 86.7 ± 7.0% | **1.75** | 2.57 | 3.10 | 3.60 |
| letter recognition | **44.9 ± 1.2%** | 37.8 ± 0.5% | **3.34** | 5.32 | 5.50 | 6.00 |
| mice | **99.9 ± 0.2%** | 71.0 ± 4.1% | **3.05** | 4.79 | **3.60** | 6.00 |
| obs network | **91.7 ± 2.1%** | 81.2 ± 1.5% | 3.48 | 3.95 | 5.30 | 5.30 |
| occupancy room | 99.4 ± 0.2% | 98.6 ± 0.4% | 4.18 | 4.77 | 5.30 | 5.60 |
| online shoppers intention | 89.3 ± 0.5% | 89.5 ± 0.6% | 3.30 | 1.84 | 5.10 | 5.60 |
| pen digits | **88.6 ± 2.0%** | 75.3 ± 1.8% | 4.76 | 4.74 | 5.80 | 6.00 |
| poker hand | 52.9 ± 1.0% | 52.0 ± 0.7% | **1.80** | 3.72 | **3.80** | 5.70 |
| sensorless | **87.4 ± 1.0%** | 74.4 ± 0.4% | **2.94** | 4.19 | 4.90 | 6.00 |
| Average | 82.3% | 76.6% | 2.93 | 3.68 | 4.69 | 5.40 |

## E.7 Comparision between SER-DT and $EC^2$

As noted in Section 5 and Section 6, SER-DT combines the theoretically best possible guarantee with very competitive performance in practice. The results shown in this section give evidence that in general optimal approximation guarantee on the (average) depth does not imply good results in terms of explanation size and accuracy.

Table 9 presents the comparison between the $EC^2$ algorithm and our SER-DT, with respect to accuracy and explanation size. $EC^2$ is a greedy algorithm proposed by [22] which is proved to achieve $O(\log n)$-approximation guarantee for the minimization of the average depth. Note that this means that it attains the best approximation guarantee that it is possible in polynomial time, under the hypothesis $P \neq NP$.

In terms of accuracy SER-DT performs significantly better than $EC^2$ on almost all datasets, and, similarly, in terms of the interpretability metrics the results also show a significant difference in favour of our algorithm. More precisely, on 14 datasets (bold-faced on columns 2 and 3) we observe a difference larger than 1% in terms of accuracies; on 13 of them, our algorithm outperforms $EC^2$ while only on 1, $EC^2$ is better. When considering the average explanation size, our algorithm is better than $EC^2$ on 14 datasets, and for 8 of them (bold-faced on column 4), it improves on the $\text{expl}_{avg}$ of $EC^2$ by at least 25%. Finally, in terms of $\text{expl}_{wc}$, on 15 datasets our algorithm is at least as good as $EC^2$, and on 5 of them the improvement is at least 25% (bold-faced in column 6).

Table 10: Test Accuracy, $\text{depth}_{avg}$ and $\text{depth}_{wc}$ for `FactorExpl` = 0.97. Each entry is the average of 10 runs using different seeds to select the examples in the training and testing set.

| Dataset | Test Accuracy | | $\text{depth}_{avg}$ | | $\text{depth}_{wc}$ | |
|---|---|---|---|---|---|---|
| | SER-DT | $EC^2$ | SER-DT | $EC^2$ | SER-DT | $EC^2$ |
| anuran | **94.8 ±0.5%** | 89.4 ± 1.7% | 5.38 | 4.83 | 6.00 | 6.00 |
| audit risk | 99.9 ±0.3% | 99.3 ± 0.8% | **1.00** | 4.34 | **1.00** | 5.90 |
| avila | **61.5 ±2.1%** | 57.7 ± 1.2% | 5.14 | 5.04 | 6.00 | 6.00 |
| banknote | **97.6 ±0.8%** | 92.3 ± 1.7% | 4.39 | 3.85 | 6.00 | 6.00 |
| bankruptcy polish | 96.6 ±0.7% | 97.4 ± 0.4% | 4.45 | 2.40 | 6.00 | 6.00 |
| cardiotocography | **89.5 ±1.3%** | 80.1 ± 3.3% | 4.98 | 5.15 | 6.00 | 6.00 |
| collins | 13.2 ±1.8% | **16.0 ± 1.2%** | 5.89 | 5.89 | 6.00 | 6.00 |
| default credit card | **82.0 ±0.4%** | 80.5 ± 0.3% | **2.15** | 3.79 | 6.00 | 6.00 |
| dry bean | **90.1 ±0.4%** | 83.9 ± 0.5% | 4.76 | 5.03 | 6.00 | 6.00 |
| eeg eye state | **74.1 ±0.5%** | 71.1 ± 0.5% | 5.15 | 5.19 | 6.00 | 6.00 |
| htru2 | 97.7 ±0.2% | 97.7 ± 0.1% | 2.80 | 3.26 | 6.00 | 6.00 |
| iris | **94.2 ±2.6%** | 86.7 ± 7.0% | **2.50** | 3.34 | 4.90 | 5.70 |
| letter recognition | **44.9 ±1.2%** | 37.8 ± 0.5% | 5.96 | 5.87 | 6.00 | 6.00 |
| mice | **99.9 ±0.2%** | 71.0 ± 4.1% | **3.05** | 4.93 | **3.60** | 6.00 |
| obs network | **91.7 ±2.1%** | 81.2 ± 1.5% | 4.47 | 4.62 | 6.00 | 6.00 |
| occupancy room | 99.4 ±0.2% | 98.6 ± 0.4% | 4.72 | 5.64 | 6.00 | 6.00 |
| online shoppers intention | 89.3 ±0.5% | 89.5 ± 0.6% | 3.89 | 3.49 | 6.00 | 6.00 |
| pen digits | **88.6 ±2.0%** | 75.3 ± 1.8% | 5.73 | 5.65 | 6.00 | 6.00 |
| poker hand | 52.9 ±1.0% | 52.0 ± 0.7% | 4.61 | 4.71 | 6.00 | 6.00 |
| sensorless | **87.4 ±1.0%** | 74.4 ± 0.4% | 5.26 | 5.47 | 6.00 | 6.00 |
| Average | 82.3% | 76.6% | 4.31 | 4.62 | 5.58 | 5.98 |

Table 10 shows the analogous results for the metrics $\text{depth}_{avg}$ and $\text{depth}_{wc}$. We bold-faced values in column 4 and 6 that show an improvements by at least 25% over $EC^2$. We notice that for 12 datasets our algorithm has better $\text{depth}_{avg}$ than $EC^2$. Moreover, considering those where SER-DT's performance is worse, on only one the difference is larger than 25% (bankruptcy polish). The values for the metric $\text{depth}_{wc}$ are similar for both algorithms, which is likely due to having set an upper bound of 6 on the maximum depth. Nonetheless, we observe that for 2 datasets the performance of SER-DT is significantly better (audit risk and mice).

### E.8 Post-pruning experiments

In this section, we present the results obtained through the application of a post-pruning strategy for trees. The algorithm consisted of checking if merging two leaves would not decrease the accuracy in a validation set formed by half of the original test set. If so, these leaves are joined and the algorithm proceeds recursively to the root.

Table 11 presents the results obtained before and after post-pruning for SER-DT, limiting the maximum depth to 6 and `FactorExpl` $= 0.97$. We observed that post-pruning did not have a significant effect in terms of accuracy, leading to differences of not more than $0.5\%$ for all datasets. In terms of explanation size, post-pruning produces smaller $\text{expl}_{avg}$ for all datasets (as expected, since we are joining nodes) and for 4 the gain was more than $20\%$ (bold-faced in table). Similarly, the pruning algorithm led to no worse values of $\text{expl}_{wc}$ for all datasets, and for 3 of these, the difference was greater than $20\%$.

Table 11: Test Accuracy, $\text{expl}_{avg}$ and $\text{expl}_{wc}$ for SER-DT with `FactorExpl` $= 0.97$, with and without post-pruning. Each entry is the average of 10 runs using different seeds to select the examples in the training and testing set. We bold-faced the values (columns 4,5,6 and 7) that represent an improvement of more than $20\%$ in terms of explainability.

| Dataset | Test Accuracy | | $\text{expl}_{avg}$ | | $\text{expl}_{wc}$ | |
|---|---|---|---|---|---|---|
| | No pruning | Pruning | No pruning | Pruning | No pruning | Pruning |
| anuran | 94.7% | 94.5% | 4.78 | 4.07 | 6.00 | 6.00 |
| audit risk | 99.9% | 99.9% | 1.00 | 1.00 | 1.00 | 1.00 |
| avila | 61.2% | 60.9% | 3.06 | 2.98 | 4.90 | 4.90 |
| banknote | 97.8% | 97.8% | 2.44 | 2.37 | 3.80 | 3.40 |
| bankruptcy polish | 96.5% | 97.3% | 2.56 | **1.45** | 5.60 | 4.50 |
| cardiotocography | 89.1% | 89.4% | 4.30 | 3.76 | 5.90 | 5.60 |
| collins | 13.1% | 13.1% | 2.13 | **1.40** | 4.40 | **3.30** |
| default credit card | 82.0% | 82.0% | 1.45 | 1.29 | 4.50 | 3.90 |
| dry bean | 90.1% | 89.9% | 3.32 | 3.17 | 5.10 | 4.70 |
| eeg eye state | 73.9% | 73.6% | 3.69 | 3.56 | 5.90 | 5.80 |
| htru2 | 97.7% | 97.8% | 1.20 | 1.09 | 4.30 | 3.70 |
| iris | 93.7% | 93.2% | 1.75 | 1.44 | 3.10 | **1.70** |
| letter recognition | 44.9% | 44.4% | 3.34 | 3.21 | 5.50 | 5.30 |
| mice | 100.0% | 100.0% | 3.05 | 3.05 | 3.60 | 3.60 |
| obs network | 92.0% | 91.6% | 3.48 | 3.35 | 5.30 | 5.30 |
| occupancy room | 99.3% | 99.3% | 4.18 | **3.22** | 5.30 | 4.70 |
| online shoppers intention | 89.3% | 89.8% | 3.30 | **1.81** | 5.10 | **4.00** |
| pen digits | 88.7% | 88.3% | 4.76 | 4.56 | 5.80 | 5.40 |
| poker hand | 52.9% | 52.9% | 1.80 | 1.78 | 3.80 | 3.80 |
| sensorless | 87.4% | 87.4% | 2.94 | 2.81 | 4.90 | 4.80 |
| Average | 82.2% | 82.1% | 2.93 | 2.57 | 4.69 | 4.27 |

Table 12 is similar to the previous table but shows metrics $\mathrm{depth}_{avg}$ and $\mathrm{depth}_{wc}$. For all datasets, there is no worsening in these metrics and for 6 of these there is an improvement of more than 20% in $\mathrm{depth}_{avg}$, while for 1 dataset there is an improvement of more than 20% in $\mathrm{depth}_{wc}$.

Tables 13 and 14 show the results for `CART` and SER-DT after applying post-pruning. We observe similar accuracy values for both algorithms. On 7 of the datasets (bold-faced in columns 2 and 3) we observe a difference of more than 1% in accuracies; on 3 of them, SER-DT outperforms `CART` while on the remaining 4, `CART` is better. In terms of $\mathrm{expl}_{avg}$ and $\mathrm{expl}_{wc}$, our algorithm is clearly better. Only for 2 datasets `CART` had better $\mathrm{expl}_{avg}$ than SER-DT and for 7 datasets our algorithm outperforms `CART` by more than 20%. Regarding $\mathrm{expl}_{wc}$ our algorithm was worst for only 3 datasets, and for 4 the improvement was more than 20% compared to `CART`.

For depth-related metrics, we noted more balanced values between the two algorithms when comparing to Table 3 (without post-pruning). As shown in Table 14, considering $\mathrm{depth}_{avg}$ our algorithm is better than `CART` for 12, worst for 6 and equal for 2 datasets. For 3 datasets the difference is more than 20%; for 2 of these SER-DT is better while for the remaining one, `CART` is the winner. In terms of $\mathrm{depth}_{wc}$ we observe close values between the algorithms: only for the dataset bankruptcy the difference is more than 20%. In this sense, we observe that dataset bankruptcy was an outlier considering explainability metrics.

Table 12: $\mathrm{depth}_{avg}$ and $\mathrm{depth}_{wc}$ for SER-DT with `FactorExpl` = 0.97, with and without post-pruning. Each entry is the average of 10 runs using different seeds to select the examples in the training and testing set. We bold-faced the values (columns 2, 3, 4 and 5) that represent an improvement of more than 20% in terms of explainability.

| Dataset | $\mathrm{depth}_{avg}$ | | $\mathrm{depth}_{wc}$ | |
|---|---|---|---|---|
| | No pruning | Pruning | No pruning | Pruning |
| anuran | 5.38 | 4.44 | 6.00 | 6.00 |
| audit risk | 1.00 | 1.00 | 1.00 | 1.00 |
| avila | 5.14 | 4.99 | 6.00 | 6.00 |
| banknote | 4.39 | 3.69 | 6.00 | 5.70 |
| bankruptcy polish | 4.45 | **2.01** | 6.00 | 5.30 |
| cardiotocography | 4.98 | 4.12 | 6.00 | 6.00 |
| collins | 5.89 | **4.38** | 6.00 | 5.90 |
| default credit card | 2.15 | 1.76 | 6.00 | 5.90 |
| dry bean | 4.76 | 4.40 | 6.00 | 6.00 |
| eeg eye state | 5.15 | 4.85 | 6.00 | 6.00 |
| htru2 | 2.80 | **2.12** | 6.00 | 5.80 |
| iris | 2.50 | **1.74** | 4.90 | **2.30** |
| letter recognition | 5.96 | 5.77 | 6.00 | 6.00 |
| mice | 3.05 | 3.05 | 3.60 | 3.60 |
| obs network | 4.47 | 4.31 | 6.00 | 6.00 |
| occupancy room | 4.72 | **3.52** | 6.00 | 6.00 |
| online shoppers intention | 3.89 | **2.19** | 6.00 | 5.90 |
| pen digits | 5.73 | 5.46 | 6.00 | 6.00 |
| poker hand | 4.61 | 4.56 | 6.00 | 6.00 |
| sensorless | 5.26 | 5.08 | 6.00 | 6.00 |
| Average | 4.31 | 3.67 | 5.58 | 5.37 |

Table 13: Test Accuracy, $\text{expl}_{avg}$ and $\text{expl}_{wc}$ for SER-DT (`FactorExpl` $= 0.97$) and `CART`, both with post-pruning. Each entry is the average of 10 runs using different seeds to select the examples in the training and testing set. We bold-faced the values that represent improvements of at least $1\%$ in accuracies (columns 2 and 3) and increase of more than $20\%$ in terms of explainability (columns 4, 5, 6 and 7).

| Dataset | Test Accuracy | | $\text{expl}_{avg}$ | | $\text{expl}_{wc}$ | |
|---|---|---|---|---|---|---|
| | CART | SER-DT | CART | SER-DT | CART | SER-DT |
| anuran | 94.3% | 94.5% | 4.58 | 4.07 | 6.00 | 6.00 |
| audit risk | 99.9% | 99.9% | 1.00 | 1.00 | 1.00 | 1.00 |
| avila | **62.7%** | 60.9% | 4.17 | **2.98** | 5.40 | 4.90 |
| banknote | 98.2% | 97.8% | 2.51 | 2.37 | 3.20 | 3.40 |
| bankruptcy polish | 97.5% | 97.3% | **1.15** | 1.45 | **2.50** | 4.50 |
| cardiotocography | 89.4% | 89.4% | 4.02 | 3.76 | 5.40 | 5.60 |
| collins | **14.5%** | 13.1% | 3.94 | **1.40** | 5.40 | **3.30** |
| default credit card | 82.1% | 82.0% | 2.38 | **1.29** | 5.10 | **3.90** |
| dry bean | 89.7% | 89.9% | 3.89 | 3.17 | 5.80 | 4.70 |
| eeg eye state | 72.9% | 73.6% | 4.18 | 3.56 | 5.80 | 5.80 |
| htru2 | 97.8% | 97.8% | 1.15 | 1.09 | 3.70 | 3.70 |
| iris | 93.7% | 93.2% | 1.54 | 1.44 | 1.80 | 1.70 |
| letter recognition | **47.9%** | 44.4% | 5.37 | **3.21** | 6.00 | 5.30 |
| mice | 100.0% | 100.0% | 3.05 | 3.05 | 3.60 | 3.60 |
| obs network | 88.4% | **91.6%** | 4.09 | 3.35 | 5.80 | 5.30 |
| occupancy room | 99.3% | 99.3% | 3.17 | 3.22 | 4.90 | 4.70 |
| online shoppers intention | 89.8% | 89.8% | 2.49 | **1.81** | 5.50 | **4.00** |
| pen digits | 86.1% | **88.3%** | 4.98 | 4.56 | 6.00 | 5.40 |
| poker hand | **55.0%** | 52.9% | 4.28 | **1.78** | 5.10 | **3.80** |
| sensorless | 80.1% | **87.4%** | 3.85 | **2.81** | 5.40 | 4.80 |
| Average | 82.0% | 82.1% | 3.29 | 2.57 | 4.67 | 4.27 |

Table 14: $\text{depth}_{avg}$ and $\text{depth}_{wc}$ for SER-DT (`FactorExpl` $= 0.97$) and `CART`, both with post-pruning. We bold-faced the values that represent an improvement of more than $20\%$ in terms of explainability.

| Dataset | $\text{depth}_{avg}$ | | $\text{depth}_{wc}$ | |
|---|---|---|---|---|
| | CART | SER-DT | CART | SER-DT |
| anuran | 4.72 | 4.44 | 6.00 | 6.00 |
| audit risk | 1.00 | 1.00 | 1.00 | 1.00 |
| avila | 5.19 | 4.99 | 6.00 | 6.00 |
| banknote | 3.74 | 3.69 | 6.00 | 5.70 |
| bankruptcy polish | **1.17** | 2.01 | **2.60** | 5.30 |
| cardiotocography | 4.19 | 4.12 | 5.70 | 6.00 |
| collins | 4.95 | 4.38 | 6.00 | 5.90 |
| default credit card | 2.39 | **1.76** | 5.10 | 5.90 |
| dry bean | 4.55 | 4.40 | 6.00 | 6.00 |
| eeg eye state | 5.22 | 4.85 | 6.00 | 6.00 |
| htru2 | 2.52 | 2.12 | 5.80 | 5.80 |
| iris | 1.71 | 1.74 | 2.10 | 2.30 |
| letter recognition | 5.80 | 5.77 | 6.00 | 6.00 |
| mice | 3.05 | 3.05 | 3.60 | 3.60 |
| obs network | 4.19 | 4.31 | 6.00 | 6.00 |
| occupancy room | 3.29 | 3.52 | 5.70 | 6.00 |
| online shoppers intention | 2.75 | **2.19** | 6.00 | 5.90 |
| pen digits | 5.41 | 5.46 | 6.00 | 6.00 |
| poker hand | 4.28 | 4.56 | 5.20 | 6.00 |
| sensorless | 5.12 | 5.08 | 6.00 | 6.00 |
| Average | 3.76 | 3.67 | 5.14 | 5.37 |