# OpenReview forum: "Decision Trees with Short Explainable Rules"
_NeurIPS.cc/2022/Conference — NeurIPS 2022 Accept_

### Official Review · Reviewer_Bn48 · 2022-07-07

**Rating:** 6
**Confidence:** 3
**Soundness:** 2 fair
**Presentation:** 3 good
**Contribution:** 2 fair

**Summary:**

The paper proposes explanation size as a novel measure to capture the interpretability of a decision tree. The explanation size of a leaf is defined as the number of distinct attributes occurring in nodes from the root to the leaf. This is an intuitively sensible notion, as it captures the size of the decision rule corresponding to the leaf. The metric is then lifted to whole trees by either considering the average or worst-case explanation size among all leafs.

As the first major contribution, the authors give a tight characterization of best possible guarantees achievable by a decision tree on the trade-off of simultaneously optimizing explanation size and the traditional criterion of tree depth. In particular, a theorem is given that states that for all datasets there are trees which have optimal explanation size and a depth within a factor of 2 of the optimal depth plus a logarithmic term in the sample size. This is complemented by a lower bound that shows that there are problem inputs for which depth and explanation size are different for all binary trees by a logarithmic factor in sample size. While not mentioned explicitly, these results seem to refer to trees that achieve zero classification error.

Moreover, the authors present a greedy top-down tree induction algorithm called Short Explainable Rules (SER-DT) that aims to find minimum explanation size trees by using a novel splitting criterion called weighted pairwise misclassification. This algorithm guarantees an O(ln n) approximation for binary decision trees of approximately optimal average / worst-case explanation size and depth, and the experiments over 20 datasets indicate that the new algorithm is competitive with the commonly used CART algorithm.

**Questions:**

Why is there no comparison between the proposed algorithm and algorithms from the literature that optimize depth? Given the demonstrated compatibility of small depth and small explanation size, this seems like a relevant baseline.

What is the significance of the example weights in the problem definition? Most commonly, we consider unweighted prediction problems, which are equivalent to using constant weights. The theorem as given allows us to choose these constant weights arbitrarily small, in effect letting the log n term vanish. This seems counterintuitive.


**Limitations:**

Besides the above mentioned limitations in terms of comparison to other tree optimisation paradigms, there is also a lack of guarantees / discussion for the tradeoff between explanation size and accuracy – once such a tradeoff is allowed by considering, as usual, no perfect classification.


**Strengths And Weaknesses:**

The introduced explanation size is a simple, apparently useful, yet, to my knowledge, novel metric to measure the interpretability of decision trees. However, note that this notion is closely related to rule sizes (as in number of queries in a rule), which is considered in the rule learning literature, e.g.,
* Ignatiev, Alexey, Filipe Pereira, Nina Narodytska, and Joao Marques-Silva. “A SAT-Based Approach to Learn Explainable Decision Sets.” In International Joint Conference on Automated Reasoning, 627–45. Springer, 2018.
* Proença, Hugo M., and Matthijs van Leeuwen. “Interpretable Multiclass Classification by MDL-Based Rule Lists.” Information Sciences 512 (February 2020): 1372–93.
* Probably has been considered in early rule learning work as well; I recommend checking the survey “Seperate-and-conquer Rule Learning” by Furnkranz for an overview of work from the 1990s.

The presented algorithm is simple yet shown theoretically and practically to achieve its goal. It would be useful though to discuss why the proposed splitting criterion based on reducing the number of discrepancy pairs leads to this result, as this as well as the other steps of the algorithm do not seem to directly relate to explanation size. In this context it would also be useful to discuss the relation to tree optimization that minimizes the total size of the tree (number of nodes), e.g.,

* Bessiere, Christian, Emmanuel Hebrard, and Barry O’Sullivan. “Minimising Decision Tree Size as Combinatorial Optimisation.” In International Conference on Principles and Practice of Constraint Programming, 173–87. Springer, 2009.

Related to that, the empirical study, while overall showing good accuracy/explanation size trade-offs relative to CAR, does not contain a comparison with a method that directly optimizes depth, which seems relevant given the theoretical results of the paper, and the cited empirical work that assesses that depth is indicative of interpretability. A further caveat is that theoretical and empirical results might relate to different variants of the algorithm: the parameter FactrorExpl seems to not play a role in the theoretical analysis. However, it is tuned for the experiments.

This relates to the clarity of the presentation, which is overall good. However, there are some missing implicit definitions and assumptions. Most importantly, the underlying problem definition seems to assume that the resulting trees have to perfectly classify the given training data. This is a reasonable definition. However, it is different from the usual ML problem, in which either tree pruning or other forms of regularization are applied, allowing the returned tree to have non-zero classification error. See for example:

* Hu, Xiyang, Cynthia Rudin, and Margo Seltzer. “Optimal Sparse Decision Trees.” In 33rd Conference on Neural Information Processing Systems, 9, 2019.

With this definition, it is also implicitly assumed that the there are perfectly classifying trees, which is not necessarily guaranteed in the usual statistical settings.

Other minor issues the term “n” is used without definition (e.g., Theorem 1), and, as mentioned, the hyper-parameter “FactorExpl”, which seems to be introduced as an afterthought in the empirical section.

---

> ### Author Response · Authors · 2022-08-02
> **Reply to Reviewer Bn48**
>
> Thank you for your comments. We are happy that you found our novel interpretability metric interesting and useful.  We also thank you for the list of pointers to additional literature that we will definitely take into consideration when preparing the final version.
> Below, we address the issues raised in your review.
>
> **QUESTIONS:**
>
> **Why is there no comparison between the proposed algorithm and algorithms from the literature that optimize depth? Given the demonstrated compatibility of small depth and small explanation size, this seems like a relevant baseline.**
>
> Reply: We assume that you refer to the algorithms proposed in the papers cited in (Lines 96-106).  We note that most of these papers do not present experiments, they are essentially purely theoretical.
>
> The main reason why we did not make this comparison is that we did not believe that a raw implementation of the algorithms in these papers would lead to competitive results with CART (or another standard algorithm) in terms of accuracy. One reason for our belief is that although our algorithm, without the use of the FactorExpl (setting it to 0), has an optimal approximation guarantee (assuming P != NP) with respect to the minimization of the depth, its accuracy in this case (FactorExpl=0) is low.  Indeed, as shown in the leftmost graph of Figure 2 its accuracy is about 8 points on average lower than the average accuracy achieved when FactorExpl is close to 1. We believe that the same poor accuracy would be seen with the raw implementation of the algorithms proposed in the literature.
>
> So in order to have fairer comparisons, in the theoretical part, we are comparing our algorithm with the algorithms that minimize the depth in terms of theoretical bounds, and on the practical side, we compare our algorithm with CART, a standard algorithm for building decision trees with good accuracy.
>
> That said, following your suggestion, we have implemented one of the methods that optimize the average depth, the EC2 algorithm from [*Near-optimal bayesian active learning with noisy observations*, Golovin et al., NIPS 2010]. When tested on the same datasets we used for our experimental analysis, it achieved average accuracy equal to 76.6% while CART and SER-DT (our algorithm) achieved more than 82%, as shown in the last line of Table 1. Moreover, in terms of average explanation size EC2 achieves 3.68 while SER-DT achieves 2.93 and CART 3.97. Finally, in terms of average depth EC2 achieves 4.62 while SER-DT achieves 4.31 and CART 4.70.
> In summary, SER-DT performs much better than EC2 on all metrics.
>
> We updated the supplementary material (Section E.7, pages 41-43) to include Tables 9 and 10 covering this experiment.
>
> **What is the significance of the example weights in the problem definition? Most commonly, we consider unweighted prediction problems, which are equivalent to using constant weights. The theorem as given allows us to choose these constant weights arbitrarily small, in effect letting the log n term vanish. This seems counterintuitive.**
>
> Reply: The inclusion of weights is very common in the literature about building optimal decision trees which is referred to in Lines 96-106. In terms of interpretability, this flexibility may be interesting since it allows to prioritize the explanation of some classes by assigning large weight to the examples in the chosen classes.
>
> Regarding the theoretical results, more precisely Theorem 1, notice that the weights are only taken into account for the average depth/explanation size, in which case the additive log term in our result is multiplied by the sum of the weights. So this makes both bounds in Theorem 1 actually invariant to rescaling the weights (and the same is true for the multiplicative guarantee of Observation 1).
>
> **OTHER ISSUES**
>
> **The presented algorithm is simple yet shown theoretically and practically to achieve its goal. It would be useful though to discuss why the proposed splitting criterion based on reducing the number of discrepancy pairs leads to this result, as this as well as the other steps of the algorithm do not seem to directly relate to explanation size.**
>
> Reply: We will add this discussion to the final version.
>
> **However, there are some missing implicit definitions and assumptions. Most importantly, the underlying problem definition seems to assume that the resulting trees have to perfectly classify the given training data**.
>
> Reply.  Indeed, on lines 125 and 126 (in the section describing the theoretical model) we explicitly say that we consider the exact classification model, namely the case where the decision tree must correctly classify all the objects in the instance, i.e. each object is required to reach some
> leaf associated to its correct class. We will nonetheless emphasize better this choice of the model in the final version of the paper.

---

### Official Review · Reviewer_RMFa · 2022-07-09

**Rating:** 7
**Confidence:** 3
**Soundness:** 3 good
**Presentation:** 2 fair
**Contribution:** 3 good

**Summary:**

The paper introduces a metric for evaluating decision trees based on explanation size. Using this metric, The authors establish theoretical bounds on identifying trees with both optimal depth and explanation size, and develop an algorithm, SER-DT, for regularizing decision trees based on explanation size. The proposed algorithm shows improvement in reducing explanation while maintaining similar performance when compared to a competing method.


**Questions:**

* Why is there such high variance in accuracy results (Table 1) between SER-DT and CART? Why is the reported accuracy is higher for SER-DT than CART for a number of datasets (e.g. dry bean, eeg eye estate)? This seems to imply that regularizing for explanation length actually increases performance.
* How was FactorExpl=0.97 chosen for the experiments? Are there any guidelines for selecting this parameter in practice?

**Limitations:**

Limitations have been adequately discussed in the paper.

**Strengths And Weaknesses:**


**Strengths:**
* The proposed definition of explanation size is natural and intuitive. Decision trees are used in a number of applications for their interpretable properties; the proposed method improves on this interpretability and therefore could be of high interest to the community.
* The paper has a good theoretical exploration of the tradeoff between tree depth and explanation size.

**Weaknesses:**
* Comparisons with Pruning algorithms. While the experiments include comparisons with CART, the paper could be improved by including comparisons with methods that reduce tree complexity, such as pruning.
* High variance of results in Table 1 for test accuracy (see questions below). It would be beneficial if the authors could clarify these points. Error bars and/or significance testing in Table 1 would be helpful.

**Miscellaneous Minor Issues / Suggestions:**
* In table 1, it would be helpful to define what the bolded results indicate in the table caption.
* Typo in Line 85: "interpretabilty"
* Typo in Line 212: "Explanaible"
* Typo in Line 302: "intrepretability"
* Grammar in Line 303: "upper and lower bound"

**Summary:**
The paper introduces a novel approach to improving decision tree interpretability. While the paper has room for improvement regarding experimental results and comparisons with pruning methods, the strong theoretical analysis makes me lean towards acceptance.

---

> ### Author Response · Authors · 2022-08-02
> **Reply to Reviewer RMFa**
>
> Thank you for your comments. We are happy that you appreciated the topic, the novel metric, and the theoretical results.
> Below, we address the issues raised in your review.
>
> **QUESTIONS**
>
> **Why is there such high variance in accuracy results (Table 1) between SER-DT and CART? Why is the reported accuracy is higher for SER-DT than CART for a number of datasets (e.g. dry bean, eeg eye estate)? This seems to imply that regularizing for explanation length actually increases performance.**
>
> Reply:  We believe that this variance (including better results for some datasets) is an intrinsic characteristic of decision trees induction rather than a weakness of our method/experiments. In fact, it is well documented in the literature that decision trees may be very sensitive to (small) perturbations in the training set and we understand that they are also (considerably) sensitive to the choice of the splitting criterion. For instance, by replacing the Gini criterion with the Entropy criterion in CART, as we did in Table 8 (p. 34 of suppl. Material), we observe the same behavior that you mentioned in terms of variance: for 7 out of 20 datasets there is a non-negligible variance (absolute difference larger than 1%) between CART with Gini (Table 8) and CART with Entropy (Table 1).
>
> Our gains in accuracy in some datasets are indeed interesting as you pointed out and might show some gain due to the regularization effect.  However, the main conclusion that we draw from our experiments is that SER-DT is much better in terms of explainability (measured by explanation size and avg depth) and competitive in terms of accuracy.
>
> **How was FactorExpl=0.97 chosen for the experiments? Are there any guidelines for selecting this parameter in practice?**
>
> Reply: The parameter FactorExpl can be used to trade off accuracy and explainability: a larger value for FactorExpl makes the algorithm favor more accurate predictions while reducing FactorExpl leads to trees that are easier to interpret (smaller explanation size). In general, the best choice depends on the user's goal and application. A user might also create a utility function that trades-off accuracy and interpretability and apply a grid search on a validation set to find a value for FactorExpl that optimizes such function.
>
> That said, our approach was to use the corpus of 20 datasets to estimate a possible default value for FactorExpl when employing our method. Based on Figure 2, and the experiments reported in Section E.2 (Suppl. Material, p. 31), where several values of FactorExpl are tested, we conclude that values in the range [0.95,0.99] provide significative gains in terms of interpretability while incurring small or no loss in terms of average accuracy. Thus, we recommend using 0.97 as a default value.
>
> **OTHER ISSUES**
>
> **Comparisons with Pruning algorithms. While the experiments include comparisons with CART, the paper could be improved by including comparisons with methods that reduce tree complexity, such as pruning.**
>
> Reply: Thanks for raising this issue, which also allows us to elaborate some more on our empirical study.
> Please, note that our experiments already include a pre-pruning strategy, consisting in limiting the depth of the tree (which we bound to 6)
>
> In terms of post-pruning strategies, we understand that a standard one consists of allowing trees of unlimited depth/size and then keep removing leaves according to some criterion that takes into account the model complexity and mainly the expected gain in accuracy (e.g. Reduced Error Pruning). The critical point is how to control the maximum depth, which is a key parameter in view of interpretability. In fact, from our experiments in the appendix (Table 6 and 7, pp. 32 and 33) one sees that without the bound on the height, the resulting trees have a maximum depth above 20 and an average depth above 10. Since these trees have, in general, higher accuracy ( explainability has its price) with respect to the ones obtained upper bounding the maximum depth (Table 1, section 6) it appears unlikely that pos-pruning will significantly reduce the depth.
>
> Another possibility is to limit the depth of the tree, as we did in Table 1 of Section 6, and then apply post-pruning to the resulting trees. We are planning to add this experiment to the final version. That said, based on our experiments with unlimited depth, we are not very optimistic that this strategy will significantly improve the explainability for both CART and SER-DT, while maintaining and/or improving the accuracy. Moreover, the post-pruning makes the method a bit more complex (eg. the necessity of a validation set and/or more hyperparameters). In any effect, post-pruning is a good attempt and it is worth trying!
>
> **Error bars and/or significance testing in Table 1 would be helpful**
>
> Reply: On page 35 of suppl. material, we present boxplots for the experiments in Table 1.
> We will emphasize it in the body of the paper.

---

> > ### Comment · Reviewer_RMFa · 2022-08-08
> > **Response**
> >
> > Thank you for your responses, especially the clarification regarding pruning algorithms. I appreciate the addition of post-pruning results in the final version, as well as the EC2 results added in the supplement. Given that my main concern was these comparisons, I have increased my score to 7.

---

> > > ### Author Response · Authors · 2022-08-08
> > > **Thanks**
> > >
> > > Thanks again for your time and your constructive criticism!

---

### Official Review · Reviewer_9pFt · 2022-07-14

**Rating:** 5
**Confidence:** 4
**Soundness:** 3 good
**Presentation:** 3 good
**Contribution:** 3 good

**Summary:**

This paper studies the interpretable decision tree by a metric called "explanation size", then presents some theoretical results about this size in relating to the tree depth in the average and worst cases.  The paper proposes a practical algorithm using adjusted Gini-index for tree splitting, based on a hyperparameter that differentiates whether using the existing split variable or using a new variable. Numerical experiments are conducted for multiple datasets.

**Questions:**

1. For the example in Section 6,  FactorExpl  is set to be 0.97. In general, how this hyperparameter should be tuned?

2. For the above hyperparameter, is there significant room to have better choices so that the explanation size drops more, while the performance may stay competitive or even better in generalization?

3. How tight are the theoretical bounds in Section 4?


**Limitations:**

The algorithm based on adjusted Gini-index seems to be a plausible approach, but the results as illustrated in Section 6 do not show significant lifting power. In Figure 2, when FactorExpl = 0.97, explanation size drops a bit dramatic, but performance also drops in a non-negligible way. It is desirable to see more convincing results with competitive performance.

**Strengths And Weaknesses:**

1. Interpretable decision tree has been an interesting topic, which is usually measured by the tree depth or the number of terminal nodes. Explanation size is a new interesting metric in measuring the splitting rule complexity.

2. This paper presents some theoretical bounds in relating explanation size to tree depth in average and worst cases. But it is not clear how tight these bounds are. (They seem to be rather loose bounds).

3. The authors propose a practical algorithm based on an adjusted Gini-index, which also sounds reasonable. It is desirable to see how effective such algorithm may construct the interpretable decision trees that are smaller explanation size while maintain competitive prediction performance. The numerical results on Section 6 are not very convincing in the performance comparison.

---

> ### Author Response · Authors · 2022-08-02
> **Reply to Reviewer 9pFt**
>
> Thank you for your comments. We are happy that you appreciated the topic and the novel metric that we introduced.
> Below, we address the issues raised in your review.
>
> **ISSUES**,
>
> **The numerical results in Section 6 are not very convincing in the performance comparison**
>
> Reply: We do not see why the numerical results are not convincing. In Table 1, for 12 out of the 20 datasets, SER-DT is at least as good as CART in terms of testing accuracy while CART is at least as good as ours in 11 datasets. Moreover, the average accuracy of our method is actually slightly larger than that of CART and the largest difference in accuracy is in our favor (87.4% for our algorithm vs 80.1% for CART on the Sensorless dataset).
> However, In terms of explanation size our algorithm gives a very significant (around  25%)  average reduction compared to CART.
>
> When FactorExpl is set to 0.99 (see Table 5 of suppl. material, page 32) the accuracy of our algorithm matches or outperforms that of CART (results on Table 1)  in 15 out of the 20 datasets; in the 5 datasets where CART wins, its advantage is at most 0.3% on 4 datasets (so negligible), and 2% on only 1 dataset. Even with this stricter setting of parameter, the explanation size of our algorithm is better or equal to that of CART in all but one dataset, and again provide significant improvement (e.g. avg explanation size 1.94 vs 4.29f or Default, and 1,89 vs 4,30 for Poker).
>
> We believe that we provide strong evidence that our method is competitive in terms of accuracy while yielding a significant advantage in terms of explainability.
>
> **This paper presents some theoretical bounds in relating explanation size to tree depth in average and worst cases. But it is not clear how tight these bounds are. (They seem to be rather loose bounds).**
>
> Reply: As explained below, replying to one of your questions, our worst-case bound of Theorem 1 is tight. Moreover, both bounds in Observation 1 and in Theorem 3  are also tight assuming P ≠ NP (see discussion in lines 249-252).
>
> **QUESTIONS**
>
> **For the example in Section 6, FactorExpl is set to be 0.97. In general, how this hyperparameter should be tuned?**
>
> Reply: The parameter FactorExpl can be used to trade-off accuracy and explainability: a larger value for FactorExpl makes the algorithm favor more accurate predictions while reducing FactorExpl leads to trees that are easier to interpret (smaller explanation size). In general, the best choice depends on the user's goal and application. A user might also create a utility function that trades-off accuracy and interpretability and apply a grid search on a validation set to find a value for FactorExpl that optimizes such function.
>
> That said, our approach was to use the corpus of 20 datasets to estimate a possible default value for FactorExpl when employing our method. Based on Figure 2, and the experiments reported in Section E.2 (Suppl, Material, page 31), where several values of FactorExpl are tested, we conclude that values in the range [0.95,0.99] provide significative gains in terms of interpretability while incurring small or no loss in terms of average accuracy. Thus, we recommend using 0.97 as a default value.
>
> **For the above hyperparameter, is there significant room to have better choices so that the explanation size drops more, while the performance may stay competitive or even better in generalization?**
>
> Reply: Yes!  As an example, for Anuran dataset, if we use FactorExpl=0.9, rather than FactorExpl=0.97, the explanation size drops from 4.78 to 3.85 (a reduction of almost 20%) while the accuracy varies just a little, dropping from 94.8% to 94.6%. For this dataset CART achieves 94.7% of accuracy and an average explanation size equal to 5.24.  These numbers were extracted from Table 1  and Table 5 (suppl. material, p. 32)
>
> More generally, Table 5 shows that setting FactorExpl=0.95 gives 32% average reduction in explanation size compared to CART with loss of only 0.35% in average accuracy.  Furthermore, by setting FactorExpl=0.9 we obtain an even bigger 38% average reduction in explanation size compared to CART with a loss of 1.1% in average accuracy
>
> **How tight are the theoretical bounds in Section 4?**
>
> Reply:  We would like to point out that the worst-case bound in Section 4, as given in Theorem 1, is indeed tight. This is made precise by the lower bound in Theorem 2, which shows the necessary trade-off one has between optimizing explanation size and depth, i.e. improving upon the bound on the depth we achieve in Theorem 1 can only be attained at the cost of a logarithmic factor loss in the explanation
> size.
>
> Moreover, our guarantee results are also tight with respect to what is possible with polynomial time algorithms. As noted in Observation 2,  Theorem 1 also implies that O(log n)-approximation simultaneously with respect to both depth and explanation size is achievable in polynomial time (which is in fact the best possible approximation under the assumption P ≠ NP).

---

### Author Response · Authors · 2022-08-02
**Supplementary Material Revised**

**TO ALL REFEREES**

We updated in OpenReview the file corresponding to the full paper (body+ supplementary material). The 13 first pages correspond to the body of our submission.  The only difference between this new file and the previous one (original submission) is
the addition of section E.7 (pages 41-43).

This section reports a comparison between our method and EC2, a method to optimize the average depth from [*Near-optimal bayesian active learning with noisy observations*, Golovin et al, NIPS 2010];  this comparison was suggested by the third referee (code Bn48)

---

### Meta-Review · Area_Chair_vYoX · 2022-08-24

**Recommendation:** Accept
**Confidence:** Certain

**Metareview:**

The paper presents an interesting approach for using decision trees in order to provide explainable classifiers

**Award:**

No

---

### Decision · Program_Chairs · 2022-09-14

Accept